# ARMH2 is a cytosolic component of CatSper crucial for sperm function

Qingqing Zhao[1,2,3,4,13], Shiyi Lin [1,2,3,4,13], Hang Kang[5,13], Yanfei Ru[3,6], Qikui Xu[2,3,4], Zijing Yu [3,7], Xiaofang Huang[8], Carlo De Rito[9], Giulia Sassi [9], Shaojie Wang[2,3,4], Shuya Sun[10], Rui Sun[11,12], Honghan Cheng[11,12], Yi Zhu [11,12], Mingxi Liu [10], Yongdeng Zhang [3,7], Min Jiang [3,6], Riccardo Percudani[9], Jean-Ju Chung [8], Xuhui Zeng [5] ✉, Zhen Yan [2,3,4] ✉ & Jianping Wu [2,3,4] ✉

Sperm capacitation and fertilization are highly regulated by Ca²⁺ signaling. CatSper, a sperm-specific calcium channel, plays a crucial role in sperm hyperactivated motility and fertility by mediating Ca²⁺ influx into sperm. Cat-Sper is the most complicated ion channel known, comprising the pore-forming CATSPER1-4 and multiple auxiliary subunits. However, our previous structural study of mouse CatSper suggests the presence of potential component(s) that remain to be identified. The identity and functional significance of the missing piece(s) of CatSper remain elusive. Here, by combining cryo-EM, mass spectrometry, AlphaFold structure prediction, and coevolutionary analysis, we identify armadillo-like helical domain containing 2 (ARMH2) as a cytosolic component of CatSper. ARMH2 forms a cytosolic ternary subcomplex with EFCAB9 and CATSPERζ, which contributes to the stable assembly of the linear arrangement of CatSper nanodomains along the sperm tail and regulates the pH and Ca²⁺ sensitivity of the channel. Loss of ARMH2 leads to compromised physiological activation of CatSper, thereby resulting in asthenozoospermia and severe subfertility. These findings show that ARMH2 is crucial for sperm function and provide fresh insights into the composition and functional regulation of CatSper. The integrated methodology employed in identifying ARMH2 also provides valuable approaches for discovering uncharacterized components in other protein complexes.

Fertilization, the fusion of sperm and egg, is a fundamental process that initiates the beginning of new life. Normal sperm function is essential for male fertility. Spermatozoa must undergo a series of processes, including sperm hyperactivation, acrosome reaction, and fusion with the egg, to achieve successful fertilization. These physiological processes are highly regulated by calcium signaling[1,2]. The cation channel of sperm (CatSper), primarily localized in the principal piece of sperm flagellar tail, serves as a pivotal sperm-specific, pH-dependent, and low voltage-dependent calcium channel[3,4]. Calcium signal transduction mediated by CatSper is essential for sperm hyper-activated motility and successful fertilization[5–8].

CatSper is the most complicated ion channel known, comprising at least thirteen components[8–16]. Unlike canonical voltage-gated calcium channels and other members of the voltage-gated ion channel superfamily[17], CatSper forms a heterotetrameric ion-conducting pore constituted by four distinct but homologous subunits: CATSPER1, 2, 3, and 4. The auxiliary subunits include the transmembrane proteins CATSPERβ, γ, δ, and ε, SLCO6C1, CATSPERη (TMEM262), and CATSPERθ (TMEM249), and the cytosolic proteins CATSPERζ and

EFCAB9. CATSPERβ, γ, δ, and ε are single transmembrane proteins with large extracellular domains that contribute to complex assembly. SLCO6C1, CATSPERη, and CATSPERθ were initially identified in our previous cryo-EM structure[16]. Notably, the discovery of the organic anion transporter SLCO6C1 as a component of CatSper reveals that CatSper is an ultracomplex composed of both a transporter and an ion channel, termed the CatSpermasome[16].

The CatSper transmembrane subunits are largely indispensable for channel functionality and sperm fertility, emphasizing their concerted contributions to complex assembly[9,11–15,18–20]. Male mice lacking transmembrane subunits[9,11,14,15,20,21], as well as humans with loss-of-function mutations in the transmembrane subunits[22–29], are completely infertile. The two small cytoplasmic proteins, CATSPERζ and EFCAB9, which are reported to be responsible for intracellular $Ca^{2+}$ and pH sensing, have been shown to have their gene disruption linked to disorganized CatSper signaling nanodomains and male subfertility, indicating their important role in the regulation of CatSper function[8,15].

Because of the importance of CatSper in sperm physiology, it has been an ideal target for treatment of related male infertility and development of non-hormonal contraceptives since the identification of its first component[9]. However, our knowledge of the composition of CatSper remains incomplete, which makes it even more challenging to overexpress and functionally reconstitute this complicated channel in vitro, hindering in-depth functional study and drug development targeting CatSper. In our previous cryo-EM study of mouse CatSper, several densities in the transmembrane and cytosolic regions remain unassigned due to moderate local resolution, suggesting that additional component(s) of CatSper could exist[16]. Supporting this idea, C2CD6 was recently reported as another potential cytosolic component that facilitates targeting and organization of CatSper in sperm flagella[30,31]. However, whether there are any other components of CatSper remains to be further explored.

In this study, we identify ARMH2 as a cytosolic component of the CatSper channel complex through an integrated strategy by combining cryo-EM, mass spectrometry, structure prediction, and bioinformatics analysis. We find that ARMH2 interacts with EFCAB9 and CATSPERζ to form an intracellular ternary subcomplex. Genetic disruption of *Armh2* results in the complete loss of the ternary subcomplex and reduces the levels of other CatSper components on the sperm membrane, thereby disrupting CatSper nanodomain organization. This leads to compromised CatSper activation in response to intracellular pH and $Ca^{2+}$ alterations, impaired sperm hyperactivated motility, and severe subfertility in male mice. Collectively, our study offers insights into the assembly and regulation of CatSper and provides a powerful methodology for identifying complex components.

## Results

### Identification of ARMH2 as an intrinsic CatSper component

The previously reported cryo-EM map of mouse CatSper exhibited limited local resolution in the cytosolic region, which was subdivided into cytosolic map 1 and cytosolic map 2, leaving several unassigned densities[16]. To facilitate reliable component assignment in this region, we further optimized the cryo-EM sample preparation and data collection steps to improve the local map quality in the following aspects: 1) Except for the previously used lacey carbon grids, we primarily employed the 2 nm carbon film-coated grids for cryo-EM sample preparation, which provide better image contrast; 2) We collected a larger dataset (Supplementary Fig. 1). Details about data acquisition and processing are provided in the Methods. The final overall map quality is comparable to that of the previously reported map (Fig. 1a).

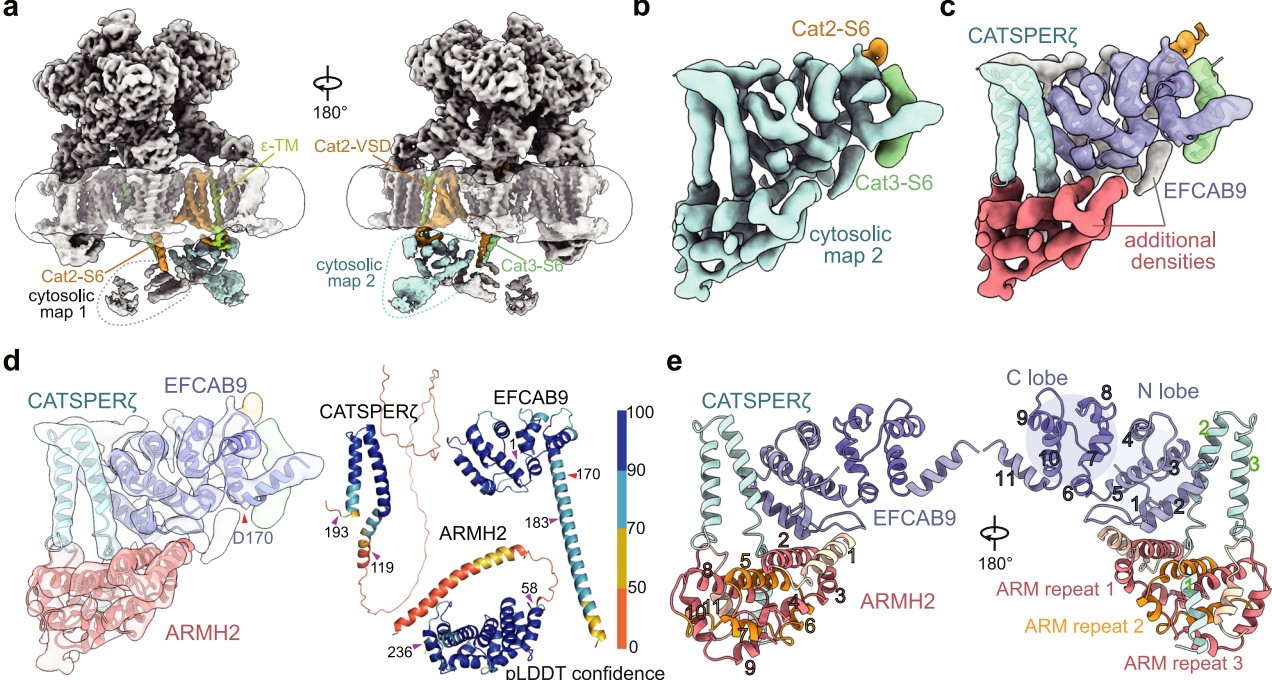

**Fig. 1 | Identification of ARMH2 as a cytosolic component of the CatSpermasome. a** Overall density map of the mouse CatSpermasome, presented in two side views. The two cytosolic regions, originally designated in our previous study[16], are indicated by dashed circles. Cytosolic map 2 is illustrated in cyan, with adjacent structural elements highlighted in color. **b** Improved map quality of cytosolic map 2 in this study. **c** Additional densities remain unassigned after modeling of EFCAB9 (slate) and CATSPERζ (cyan) into cytosolic map 2. The α helical-shaped densities below EFCAB9 are colored salmon, and the remaining densities attached to EFCAB9 are colored gray. **d** Left: AlphaFold 2 predicted structure of ARMH2 (salmon) fits perfectly into the unassigned region of cytosolic map 2 with minor adjustments. Right: The pLDDT confidence map of the AlphaFold 3-predicted structures of the ARMH2-EFCAB9-CATSPERζ ternary complex. For visual clarity, the three proteins are presented separately. The boundaries of each protein that are resolved in the cryo-EM map are indicated by purple arrows. The C-terminal helix of EFCAB9, predicted to form a long α helix, is partially resolved and kinked at Asp170 (indicated by red arrows). **e** Overall structure of the ternary subcomplex. The α helical segments of each component and the three armadillo (ARM) repeats of ARMH2 are labeled.

Notably, further improvement of map quality in the region corresponding to cytosolic map 2 by local classification and refinement yielded a local map at 4.8 Å resolution, which displayed clear secondary structural features, primarily α helices (Fig. 1b). During data processing and model building, we noticed a distinct additional density near the previously characterized EFCAB9 and CATSPERζ, which may represent an extra component of CatSper (Fig. 1c). Unfortunately, the resolution of the local map remains insufficient for clear de novo side chain assignment (Fig. 1c).

To facilitate reliable component assignment of this density, we analyzed the predicted structures by AlphaFold 2[32] of all potential candidates that are co-purified with the native CatSper and detected in our previous mass spectrometry result[16]. We found that the main body of the predicted structure of ARMH2, which contains three armadillo repeats, can be perfectly fitted into the density (Fig. 1d). Moreover, we also tried complex prediction of the candidate proteins with EFCAB9 and CATSPERζ using AlphaFold 3[33], and the interface predicted template modeling (ipTM) score of ARMH2 stands out among all other candidates, which is the only one above 0.7 (Fig. 2a). The predicted ARMH2-EFCAB9-CATSPERζ ternary subcomplex is highly consistent with the cryo-EM map (Fig. 2b). The cryo-EM resolved regions of the ARMH2-EFCAB9-CATSPERζ subcomplex are primarily in areas with high predicted local distance difference test (pLDDT) scores, supporting the assertion that the unassigned density belongs to ARMH2 (Fig. 1d). In contrast, the predicted structures of other candidates in complex with EFCAB9 and CATSPERζ exhibit lower ipTM scores and cannot fit well with the cryo-EM density (Fig. 2c).

The ARMH family contains four members, ARMH1-4, which share relatively low sequence identity among each other (< 20%). Among them, ARMH1 and ARMH2 are primarily expressed in the testis, whose functions are previously uncharacterized, whereas ARMH3 (also named C10orf76) and ARMH4 are more widely expressed, and their reported functions are not related to fertilization process[34,35]. Unlike its name indicates, ARMH4 does not contain armadillo repeats, and ARMH1 and ARMH3 contain more armadillo repeats than ARMH2, which make them unfit to the cryo-EM density in size (Supplementary Fig. 2a). Besides, structure predictions of ARMH1/3/4 in complex with CATSPERζ and EFCAB9 exhibit much lower ipTM scores compared to that of ARMH2 (Supplementary Fig. 2a), suggesting ARMH2 is the only member among ARMH members to be an intrinsic CatSper component.

## ARMH2 forms a ternary complex with EFCAB9 and CATSPERζ

Based on the cryo-EM data and structure prediction analyses, ARMH2 forms a stable ternary subcomplex with EFCAB9 and CATSPERζ (Figs. 1d, 2b). Although the resolution of the cryo-EM map does not allow for side chain assignment, we tentatively analyzed the interactions between ARMH2 and the other two components based on the predicted structure (Fig. 1e, Supplementary Fig. 3a). Notably, the residues that potentially mediate specific interactions between ARMH2 and the other two components are highly conserved across mammalian species, suggesting a conserved interaction mode of the subcomplex (Supplementary Fig. 3b–e). Consistently, structure predictions of ARMH2 in complex with EFCAB9 and CATSPERζ in various species all exhibit similar overall structures with ipTM scores higher than 0.75 (Supplementary Fig. 2b), suggesting this cytosolic subcomplex of CatSper is also conserved across non-mammalian species.

To further experimentally validate the ternary subcomplex, we attempted an in vitro pull-down assay. However, these three proteins were not well expressed individually in the HEK293 expression system, precluding the use of individually expressed proteins to verify their mutual interactions. Alternatively, we transiently co-expressed ARMH2-3×FLAG, Twin-Strep-EFCAB9, and CATSPERζ-GFP in HEK293 cells and examined whether purification of one component would pull-down the other two proteins (Fig. 2d). Notably, affinity purification of

ARMH2 using anti-FLAG resin co-purified both EFCAB9 and CATSPERζ (Fig. 2e). Similarly, affinity purification of EFCAB9 using Strep beads also pulled down ARMH2 and CATSPERζ, supporting the formation of a subcomplex between ARMH2, EFCAB9, and CATSPERζ (Fig. 2f). As a control, when only EFCAB9 and CATSPERζ were co-expressed, they could not be pulled down by anti-FLAG resin. Interestingly, although Strep affinity purification of EFCAB9 still pulled down CATSPERζ in the control group, the CATSPERζ-to-EFCAB9 ratio was lower than in the group co-expressing all three components (Fig. 2f). These results suggest that ARMH2 enhances the stability of the interaction between EFCAB9 and CATSPERζ, which is consistent with the structural analysis.

## ARMH2 co-evolves with other CatSper components

Most of the genes encoding known CatSper subunits exhibit gene coevolution patterns with simultaneous gain and loss across species. A previous genomic screen aimed at clustering genes with CatSper genes led to the identification of CATSPERθ independent of structural characterization, underscoring the power of bioinformatics analysis in identifying CatSper components[21]. In this study, we refined the CatSper coevolutionary network by clustering significant pairwise association among 65,201 gene groups from 1952 species. Using a sensitive, Hidden Markov Model (HMM)-based homology search, the Armh2 gene was found to cluster together with most CatSper genes, including Catsper1-4, Catsperb-e, and Efcab9 (Fig. 3). Reciprocally, the 10 most significant coevolutionary genes of Armh2 are all known CatSper genes (Supplementary Fig. 4). These bioinformatics analyses, in accordance with the structural and biochemical data, further support ARMH2 as an intrinsic component of CatSper.

## Armh2-null sperm exhibit lower CatSper expression levels

To better understand the functional role of ARMH2 in CatSper assembly and sperm function, we generated Armh2 knockout mice by removing both exons of the Armh2 gene using CRISPR/Cas9 genome editing (Supplementary Fig. 5a–c). Armh2 knockout male mice exhibited a normal overall phenotype, including survival, growth, and appearance. As Armh2 is a testis-specific gene, we also evaluated the weight of testes, sperm count, sperm morphology, and the histology of seminiferous tubules. No significant differences were observed between Armh2 knockout and wild-type (WT) male mice in these aspects, suggesting the absence of ARMH2 does not affect testicular development and spermatogenesis processes (Supplementary Fig. 5d–f).

Previous studies have suggested the CatSper components exhibit interdependency of protein levels in spermatozoa[8,14,15,21]. Consistent with this pattern, ARMH2 was nearly undetectable in Catsper1−/−, Efcab9−/−, and C2cd6−/− mice sperm (Fig. 4a, b). We next examined whether the absence of ARMH2 would also affect the protein levels of other CatSper components. Indeed, immunoblotting analysis of the Armh2-null sperm revealed that the detected transmembrane subunits, including CATSPER1, CATSPERβ, δ, γ, ε, and θ, were all reduced to approximately half or less compared to WT, whereas EFCAB9 and CATSPERζ were entirely absent (Fig. 4c, d). However, the extent of the effects observed in Armh2 knockout sperm differs from that in Catsperq (encoding the transmembrane component CATSPERθ) knockout sperm, where all CatSper components are undetectable (Fig. 4c). In contrast, the results are more consistent with the phenotypes observed in Efcab9 or Catsperz knockout sperm[8,15].

We further performed a quantitative proteomic analysis to systematically evaluate the changes in the CatSper components between Armh2-null and WT sperm. ARMH2, also known as 1700016G14Rik in mice and C6orf229 in humans, can be identified in proteomic databases using these alternative names[36]. Corroborating the immunoblotting results, ARMH2, EFCAB9, and CATSPERζ were undetectable in the Armh2-null sperm samples compared to WT, indicating a complete

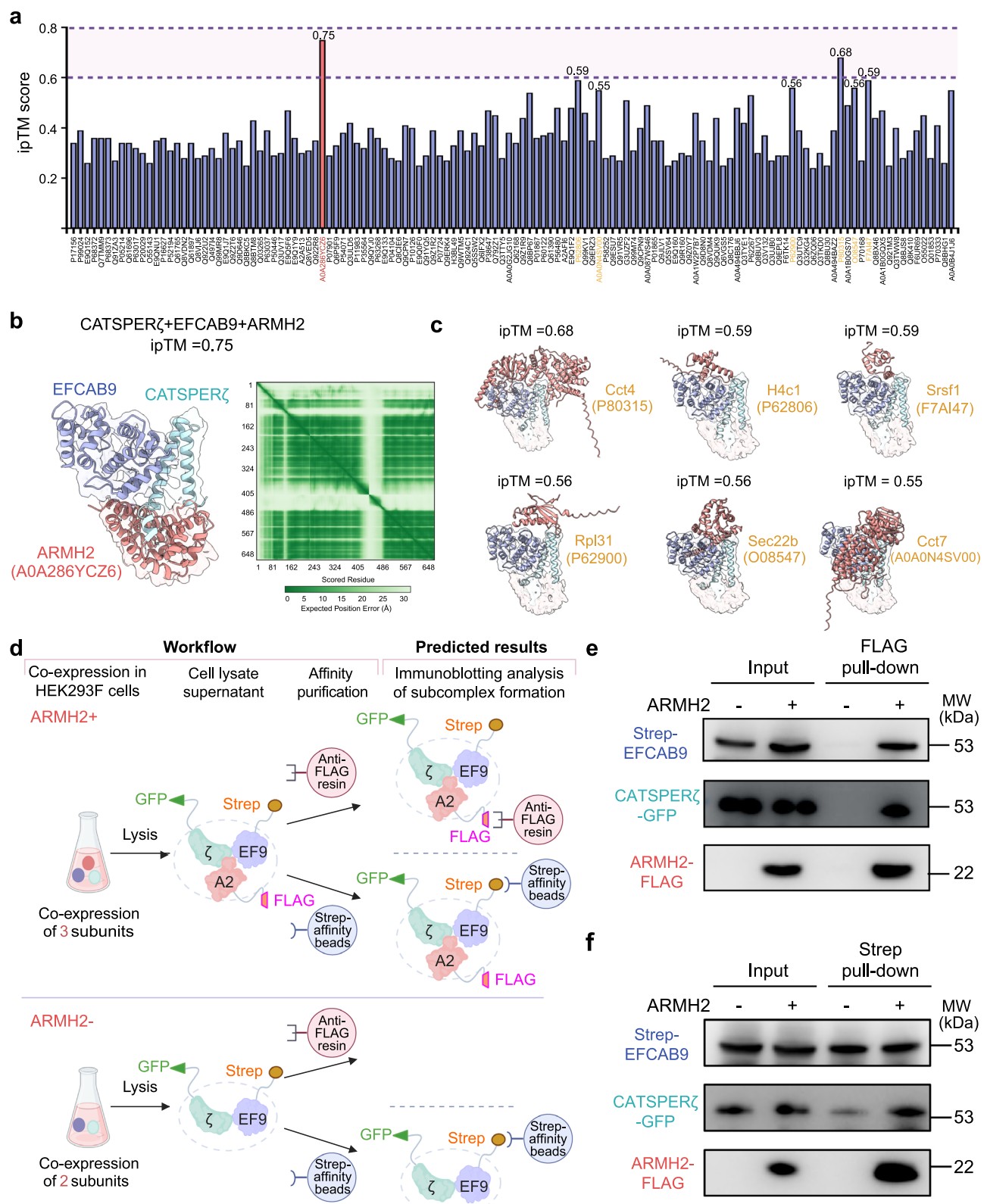

loss of EFCAB9 and CATSPERζ in *Armh2*-null sperm (Fig. 4e, Supplementary Fig. 6, and Supplementary Data 1). In contrast, all transmembrane subunits, including the pore-forming subunits CATSPER1-4 and the auxiliary subunits CATSPERβ, γ, δ, ε, θ, and SLCO6C1, were detectable in the *Armh2*-null sperm but exhibited a decreased expression pattern compared to WT (Fig. 4e, Supplementary Fig. 6, and Supplementary Data 1). These results suggest that the three

components of the cytosolic ARMH2-EFCAB9-CATSPERζ subcomplex are interdependent, yet their assembly or stability is largely independent of other CatSper components.

## ARMH2 contributes to CatSper nanodomain assembly

The CatSper channel complex organizes into specialized longitudinal $Ca^{2+}$ signaling nanodomains, arranged in four linear quadrants along

**Fig. 2 | AlphaFold prediction of ARMH2 as a component of the CatSper ternary subcomplex and biochemical validation. a** Bar plot of the ipTM scores of the predicted structures of the candidates in complex with EFCAB9 and CATSPERζ using AlphaFold 3. The UniProt IDs of the candidates are labeled. ARMH2 is highlighted in red. **b** AlphaFold 3 predicted structure of ARMH2 in complex with CATSPERζ and EFCAB9, fitted within the context of the cryo-EM density (presented in transparency). The ipTM score and the predicted aligned error is provided. **c** AlphaFold 3 predicted structures of the other six candidates (with ipTM scores above 0.55, labeled yellow in panel (**a**)) in complex with CATSPERζ and EFCAB9, fitted within the context of the cryo-EM map. **d** The workflow of a co-expression and pull-down assay to test the subcomplex interaction. EFCAB9 and CATSPERζ were co-expressed with (upper) or without (lower) ARMH2 in HEK293F cells and subsequently lysed. The supernatant was loaded onto anti-FLAG resin or Strep affinity beads for a pull-down assay. A2: ARMH2; EF9: EFCAB9; ζ: CATSPERζ. Created in BioRender. Wu, J. (2025) https://BioRender.com/x55p898. **e** Immunoblotting detections of the elution sample after affinity purification by anti-FLAG resin. **f** Immunoblotting detections of the elution sample after affinity purification by Strep affinity beads. The experiments in (**e**, **f**) were independently repeated at least three times with similar results. Uncropped blots are provided in the Source Data file.

the principal piece of the sperm flagellum[1]. Each linear CatSper nanodomain appears as two parallel lines in super-resolution imaging[8], and cryo-electron tomography (cryo-ET) reveals that each line is composed of repeating CatSper units arranged in a zigzag pattern along the length of the sperm flagellum[37]. Although it remains unclear how the quadrilinear CatSper nanodomains are formed, their intact assembly has been shown to be critical for the regulation and maintenance of hyperactivated sperm motility and swim path chirality[8,15,38]. To test whether the absence of ARMH2 disrupts CatSper nanodomain assembly, we employed 4Pi single-molecule switching nanoscopy (4Pi-SMSN) to visualize the CatSper nanodomains. A crossed mouse strain ($Armh2^{+/-}$; $Catsper1^{GFP/GFP}$) was used for the super-resolution imaging, allowing the CatSper complex to be labeled by an anti-GFP antibody, with the $Catsper1^{GFP/GFP}$ knock-in mouse strain serving as the control. Consistent with previous studies[8,15], the intact quadrilinear CatSper nanodomains are formed along the principal piece of the sperm flagellum in the $Catsper1^{GFP/GFP}$ mice strain (Fig. 4f). However, in the sperm from $Armh2^{-/-}$; $Catsper1^{GFP/GFP}$ mice, although the overall quadrilinear structure is preserved, the CatSper nanodomains are fragmented and primarily organized in a single line within each linear nanodomain (Fig. 4f and Supplementary Movie 1). Consequently, the overall CATSPER1-GFP signal in $Armh2^{-/-}$, $Catsper1^{GFP/GFP}$ sperm is visually lower than control, which is consistent with the immunoblotting and quantitative mass spectrometry results. The imaging results strongly suggest that ARMH2 is critical for the continuous linear CatSper nanodomain assembly. Notably, a similar phenomenon of disrupted quadrilinear CatSper nanodomains was observed in sperm from $Efcab9^{-/-}$, $Catsperz^{-/-}$, or $Catsperz^{-/-}$; $Efcab9^{-/-}$ double-knockout mouse strains[8,15], suggesting that ARMH2 works together with EFCAB9 and CATSPERζ in reinforcing the assembly of CatSper nanodomains.

### Knockout of ARMH2 affects sperm CatSper currents

As the absence of ARMH2 affects the overall CatSper protein levels and CatSper nanodomain assembly in sperm, we speculate that the sperm CatSper current ($I_{CatSper}$) may also be affected in $Armh2^{-/-}$ sperm. We therefore measured the $I_{CatSper}$ of the intact sperm from both $Armh2^{+/-}$ and $Armh2^{-/-}$ mice, following a previous protocol[8,39,40]. $Armh2^{+/-}$ sperm were used as controls as they exhibited $I_{CatSper}$ levels comparable to those of WT sperm. At an alkaline intracellular pH (pH$_i$ = 7.4), the inward current of $I_{CatSper}$ was recorded at $-71.17 \pm 9.19$ pA/pF at a membrane potential of $-80$ mV in the absence of Ca$^{2+}$ in $Armh2^{+/-}$ sperm. This current was increased to $-85.96 \pm 10.90$ pA/pF with 0.1 μM [Ca$^{2+}$]$_i$ and to $-97.28 \pm 8.27$ pA/pF with 10 μM [Ca$^{2+}$]$_i$, demonstrating a Ca$^{2+}$ dose-dependent manner consistent with the $I_{CatSper}$ recordings in WT sperm in a previous study (Fig. 5a, b, Supplementary Fig. 7)[8]. By contrast, in the $Armh2^{-/-}$ sperm, the inward current of $I_{CatSper}$ was measured at $-42.32 \pm 9.43$ pA/pF at a membrane potential of $-80$ mV in the absence of Ca$^{2+}$, which was significantly less than that recorded from $Armh2^{+/-}$ sperm, and the current did not progressively increase with rising Ca$^{2+}$ concentrations (Fig. 5a, b, Supplementary Fig. 7). These results suggest that the absence of ARMH2 not only reduces the amplitude of $I_{CatSper}$, but also impairs the Ca$^{2+}$ sensitivity of CatSper, likely due to a consequence of EFCAB9 loss within the cytosolic subcomplex[8].

We next examined the effect of ARMH2 knockout on the pH sensitivity of $I_{CatSper}$. For this purpose, we firstly measured the current at a lower internal buffer pH of 6.0, and then measured the current again after bath application of 10 mM NH$_4$Cl to induce intracellular alkalinization, following a previous protocol[5] (Fig. 5c). At pH$_i$ = 6.0, the $I_{CatSper}$ in both $Armh2^{+/-}$ and $Armh2^{-/-}$ sperm exhibited minimal and comparable inward conduction (Fig. 5d, e). Upon intracellular alkalinization, the $I_{CatSper}$ in $Armh2^{+/-}$ sperm increased by a fold change of 2.0-3.1, whereas in $Armh2^{-/-}$ sperm the activation was reduced, with a fold change of 1.3-1.5. This difference was particularly pronounced in the absence of Ca$^{2+}$ (Fig. 5f). These results suggest that the absence of ARMH2 also impairs the pH sensitivity of CatSper.

### Armh2-null males are severely subfertile

Dysfunction of the CatSper channel complex has been reported to be related to male infertility or subfertility in both humans and mice[8,9,11,14,15,20–29]. We have shown that the absence of ARMH2 results in reduced sperm CatSper protein levels, misalignment of the CatSper nanodomain, and altered channel gating properties. Consequently, it is expected that the sperm motility and fertility of $Armh2$-null male mice are likely impaired. Although computer-assisted sperm analysis (CASA) revealed only a slight, statistically insignificant reduction in total and progressive motility of $Armh2^{-/-}$ sperm compared to WT, significant differences were observed in hyperactivated motility parameters, such as curvilinear velocity (VCL) and amplitude of lateral head (ALH), between $Armh2^{-/-}$ and WT sperm (Fig. 6a, b, Supplementary Fig. 8a). Specifically, both VCL and ALH of WT sperm exhibited a progressive increase during incubating in Human Tubal Fluid (HTF) capacitation buffer for 90 min, suggesting successful sperm hyperactivation. In contrast, $Armh2^{-/-}$ sperm showed no significant change in VCL or ALH under the same condition, indicating a defect in hyperactivation and impaired capacitation (Fig. 6b, Supplementary Fig. 8a). Consistently, the flagellar beating waveform of $Armh2^{-/-}$ sperm was also altered. While WT sperm exhibited hyperactivated motility with increased beating amplitude following capacitation, $Armh2^{-/-}$ sperm displayed small swing amplitude and showed no significant change under capacitating condition, indicating an abnormal hyperactivation process (Fig. 6c, Supplementary Movie 2). Moreover, although the $Armh2^{-/-}$ male mice exhibited a normal overall phenotype, including normal mating behavior, their fertility rate was significantly reduced compared to WT or $Armh2^{+/-}$ male mice (Fig. 6d). Among the 16 $Armh2^{-/-}$ male mice tested in the mating experiment, only one knockout mouse could sire two pups, while all other knockout male mice were completely infertile (Fig. 6d). In vitro fertilization assays also showed that the ability of $Armh2^{-/-}$ sperm to fertilize oocytes, as indicated by the percentage of 2-cell embryos, was significantly reduced compared with WT or $Armh2^{+/-}$ sperm, decreasing from approximately 80% to 20% (Fig. 6e). Notably, embryo development analysis revealed that less than 10% of 2-cell embryos from the $Armh2^{-/-}$ group developed to the blastocyst stage, much lower than the ~70% observed in the WT group (Fig. 6f, Supplementary Fig. 8c). This indicates that the actual successful rate of in vitro fertilization using $Armh2^{-/-}$ sperm is very low, with most of the observed 2-cell embryos likely arising from parthenogenetic activation. Taken together, our findings demonstrate that

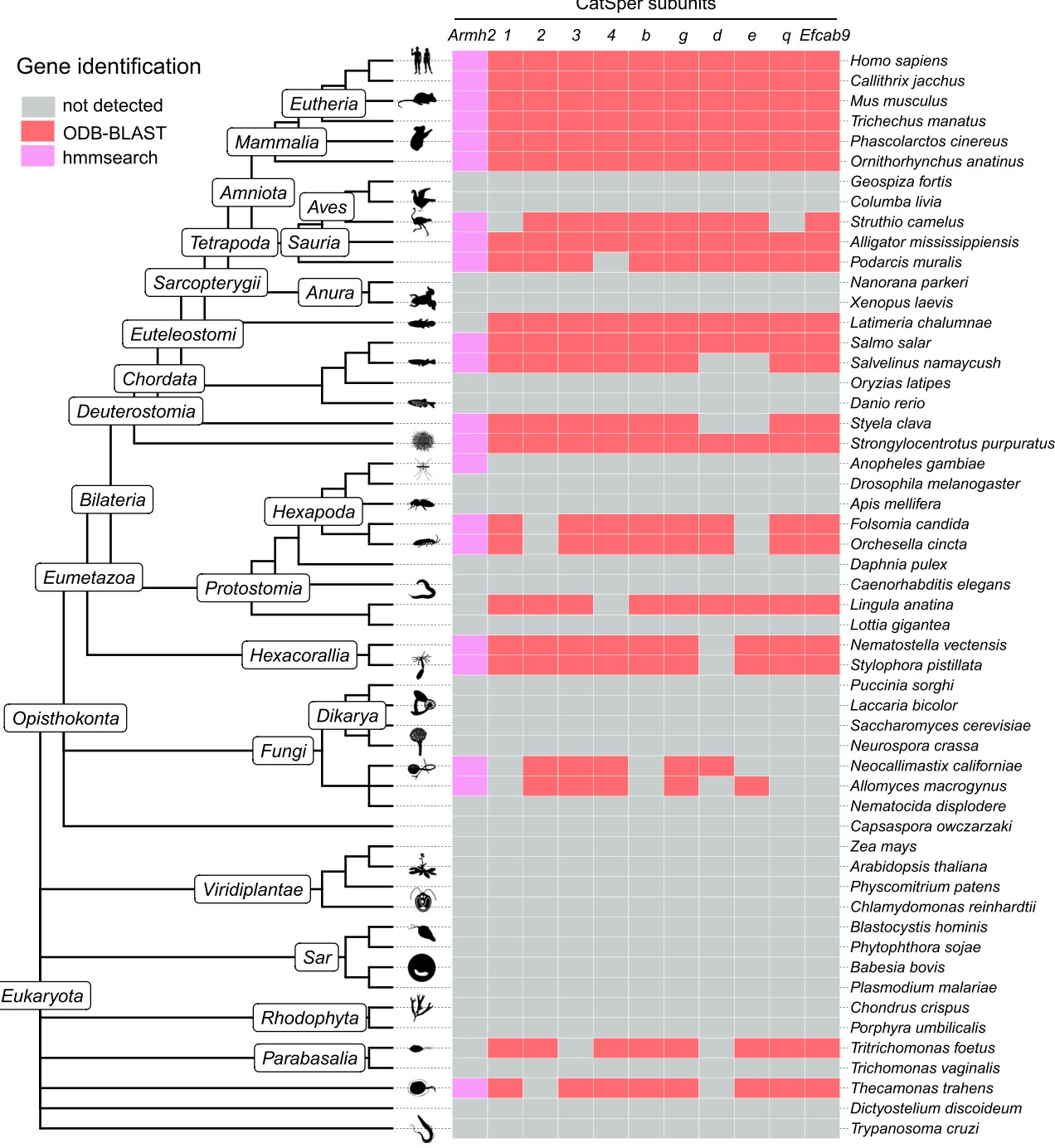

**Fig. 3 | Distribution map of *Armh2* and other CatSper subunit genes across eukaryotes.** The phylogenetic tree follows NCBI taxonomy, representing evolutionary relationships. *Armh2* distribution was identified by HMM analyses (hmmsearch), while distributions for other CatSper subunits were based on OrthoDB V.11 (ODB-BLAST). Displayed are 52 representative species selected from a broader dataset comprising 1952 eukaryotic species examined in this study. Source data are provided in the Source Data file.

the *Armh2*-null male mice are severely subfertile due to impaired sperm hyperactivation.

## Discussion

### An integrated strategy to identify ARMH2 as a CatSper component

As the most complicated ion channel, the identification of the components of CatSper has been an exciting scientific journey. Over the past two decades, genetic, biochemical, and bioinformatical approaches have successfully identified many CatSper components[41,42]. Our

previous cryo-EM study further revealed the unique power of structural biology in identifying previously unresolved CatSper components, especially small transmembrane proteins that are prone to being overlooked by other methods[16]. However, direct component identification solely based on structural studies typically requires near-atomic resolution to allow for clear side chain assignment and unambiguous sequence matching. In practice, achieving high resolution in every region of a cryo-EM map is often challenging. For large protein complexes like CatSper, densities are generally less well resolved in peripheral regions compared to the central region[16]. Consequently,

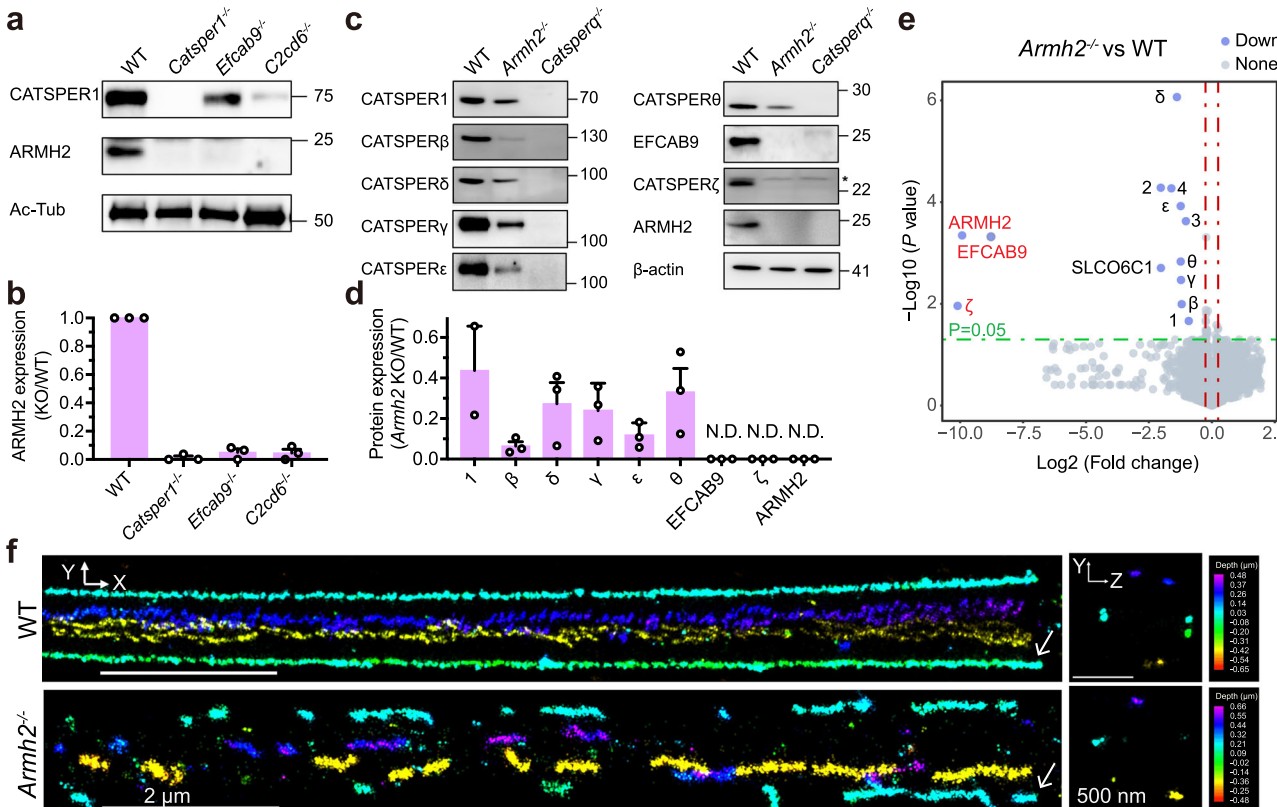

**Fig. 4 | ARMH2 contributes to CatSper expression and linear nanodomain assembly. a** Immunoblotting detections of ARMH2 and CATSPER1 in WT, *Catsper1⁻/⁻*, *Efcab9⁻/⁻* and *C2cd6⁻/⁻* sperm. Ac-Tub: Acetylated α-tubulin. This experiment was independently repeated three times with similar results. Uncropped blots are provided in the Source Data file. **b** Relative quantification analysis of the detected ARMH2 expression levels in WT and KO sperm, as shown in (**a**). Data are presented as mean ± SEM; *n* = 3, where n represents the number of biologically independent replicates. **c** Immunoblotting detections of representative CatSper subunits in WT and *Armh2⁻/⁻* sperm. *Catsperq⁻/⁻* sperm was used as a negative control. *represents a nonspecific band. The experiment was independently repeated twice for CATSPER1 and three times for the other groups, with similar results. Uncropped blots are provided in the Source Data file. **d** Relative quantification analysis of the detected CatSper components between the *Armh2⁻/⁻* and WT sperm,

as shown in (**c**). Data are represented as mean ± SEM; *n* = 3 for each group, except for CATSPER1 with *n* = 2. N.D., not detected. **e** Volcano map of quantitative mass spectrometry analysis of whole sperm cell from WT and *Armh2⁻/⁻* mice, *n* = 4 for each group. All the down-regulated CatSper subunits were labeled with purple round dots. ARMH2, EFCAB9, and CATSPERζ were not detected in *Armh2⁻/⁻* sperm and were highlighted in red. The green dash line indicates *P* = 0.05, and two red dash lines indicate Log2 (Fold change) = ±0.25. *P*-values were determined using two-tailed unpaired Student's *t*-tests. Source data are provided in the Source Data file. **f** 4Pi-SMSN images of mouse linear CatSper nanodomain in WT and *Armh2⁻/⁻* sperm. A x-y cross-section (left) and a y-z projection (right) are presented. The color bar encodes the relative distance from the focal plane along the z-axis. The white arrows indicate the annulus site between the midpiece and principal piece.

residual densities with relatively low local resolutions often remain unassigned in the cryo-EM maps of protein complexes, and identifying these densities presents significant technical challenges[16,43–45].

In this study, we identify ARMH2 as a previously uncharacterized cytosolic component of the CatSper complex, revealing that CatSper is even more complicated than previously thought. Although we have significantly improved the map quality of the cytosolic map 2 of CatSper compared to the previous study[16], the best cryo-EM density in this region only allows for secondary structural assignment (Fig. 1c). Here, we combined cryo-EM with mass spectrometry, AlphaFold structure prediction, and co-evolutionary analysis to identify the unknown cytosolic density of CatSper as ARMH2. Biochemical and functional characterizations further validate this assignment. The integrated strategy employed in this study not only adds a crucial piece to our understanding of the CatSper components but also provides a valuable reference for the identification of components of other protein complexes.

**A cytosolic subcomplex critical for channel gating**
Our data strongly support that ARMH2 forms a stable ternary subcomplex with EFCAB9 and CATSPERζ. Knockout of each of these three components results in complete loss of the other two components, but

only decreased levels of other CatSper components, indicating that the interdependence among the three cytosolic components is much higher than their interdependence with other components (Fig. 4c–e)[8,15]. This explains the overall similar phenotypes in the knockout mice of the three components, including expression levels of CatSper components, CatSper nanodomain assembly, electrophysiological properties of $I_{CatSper}$, sperm motility, and male fertility[8,15].

Our super-resolution imaging and electrophysiology data demonstrate that the CatSper channel complex remains assembled at the sperm tail, maintaining basal ion permeability, even in the absence of the ternary cytosolic subcomplex. This suggests that the subcomplex is not essential for the functional assembly of CatSper complex units. However, the disrupted CatSper nanodomain observed in *Armh2⁻/⁻* sperm indicates that the cytosolic subcomplex contributes to the stability of the higher-order linear assembly of the CatSper nanodomain (Fig. 4f). A previous cryo-ET study revealed a zigzag arrangement of the CatSper nanodomain, in which adjacent units interact via their extracellular domains and are intracellularly connected in pairs[38]. The cytosolic ternary subcomplex is likely involved in mediating the intracellular connections. Disruption of these intracellular linkages, resulting from knockout of any of the three subcomplex components, likely compromises the structural integrity of the CatSper

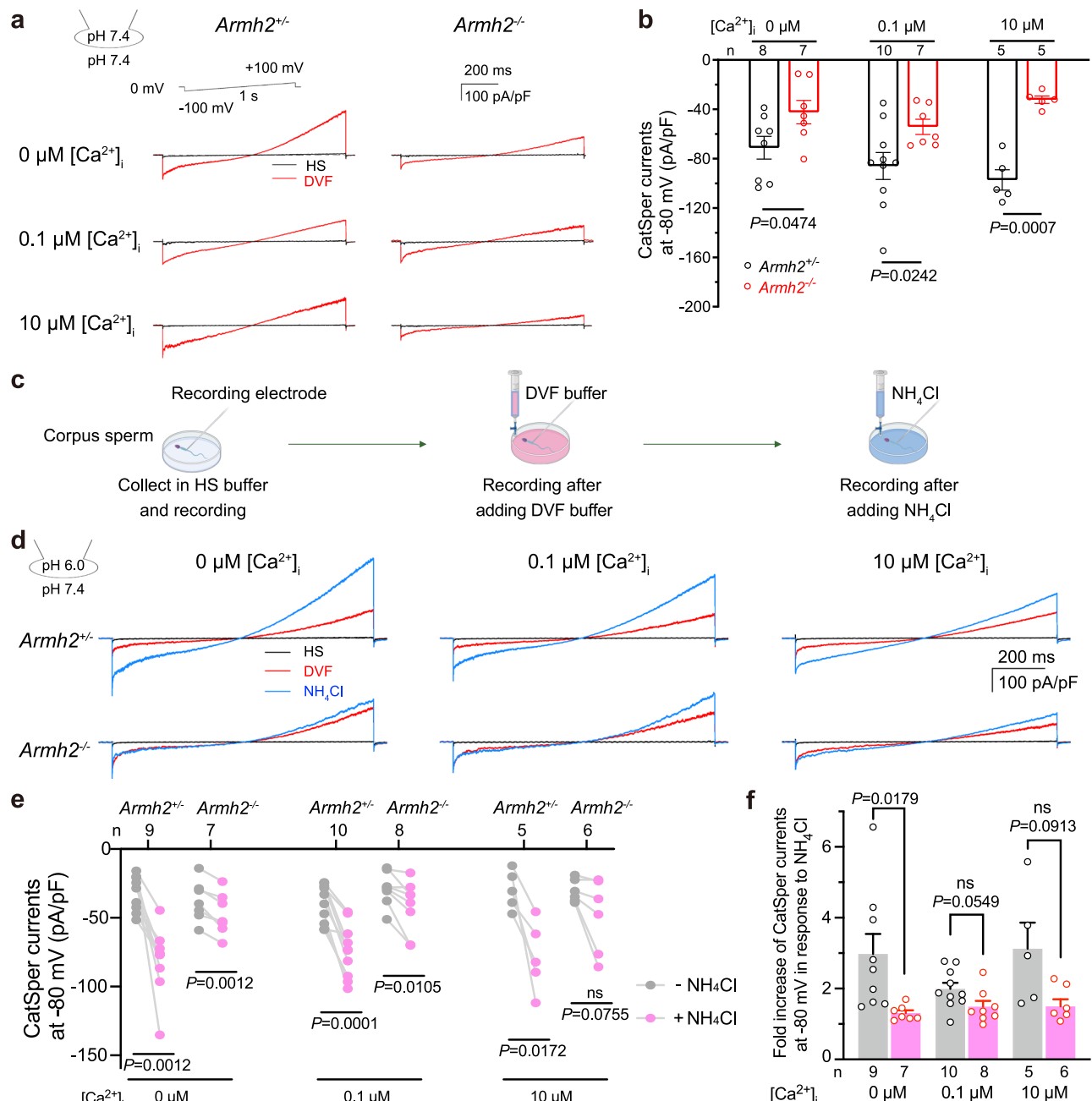

**Fig. 5 | ARMH2 knockout affects intracellular Ca²⁺ and pH sensing in CatSper.**
**a** Representative ramp current traces of the CatSper currents from *Armh2*⁺ᐟ⁻ (left) and *Armh2*⁻ᐟ⁻ (right) corpus sperm under HS (black curve) or DVF (red curve) extracellular buffer conditions. The pipette solution pH was 7.4 with 0 μM, 0.1 μM, or 10 μM free Ca²⁺. The ramp voltage stimulus was set from −100 mV to +100 mV for 1 s. HS: HEPES-buffered saline; DVF: divalent-free solution. Each experiment was independently repeated at least five times with similar results. **b** Quantitative analysis of the ramp currents at −80 mV, as recorded in (**a**). *Armh2*⁺ᐟ⁻, black; *Armh2*⁻ᐟ⁻, red. The number of biologically independent sperms (n) and the *P*-values for each group are indicated in the figure. **c** The workflow of electrophysiological recordings of $I_{CatSper}$ under different pH buffer conditions. Created in BioRender. Wu, J. (2025) https://BioRender.com/v40d041. **d** Representative ramp current traces of $I_{CatSper}$ recorded from *Armh2*⁺ᐟ⁻ and

*Armh2*⁻ᐟ⁻ corpus sperm under different intracellular free Ca²⁺ concentrations at intracellular pH 6.0, before (red curve) and after (blue curve) 10 mM NH₄Cl treatment. This experiment was independently repeated five times with similar results. **e** Quantitative analysis of the ramp currents at −80 mV recorded in (**d**). Data are presented as paired line graphs. The number of biologically independent sperms (n) and *P*-values for each group are indicated in the figure. ns: no significant difference. **f** Fold increase of the inward $I_{CatSper}$ at −80 mV after intracellular alkalization by NH₄Cl treatment in *Armh2*⁺ᐟ⁻ (gray) and *Armh2*⁻ᐟ⁻ (magenta) sperm. Data are presented as mean ± SEM. The number of biologically independent replicates (n) and *P*-values for each group are indicated in the figure. *P*-values were determined using two-tailed unpaired Student's *t*-tests in (**b**, **f**) and two-tailed paired Student's *t*-tests in (**e**). Source data are provided in the Source Data file.

nanodomain, yet does not prevent its formation. This may account for the intermittent occurrence of the CatSper nanodomain observed in sperm from *Armh2* knockout mice (Fig. 4f, Supplementary Movie 1).

In addition to its potential role in contributing to CatSper nanodomain formation, the cytosolic subcomplex plays a crucial role in

regulating the gating of the CatSper channel by sensing of intracellular Ca²⁺ and alkalinization. Our data demonstrate that the absence of ARMH2 affects the gating properties of the CatSper channel in response to intracellular Ca²⁺ and alkalinization, aligning with previous findings on EFCAB9 and CATSPERζ[8,15]. These results collectively

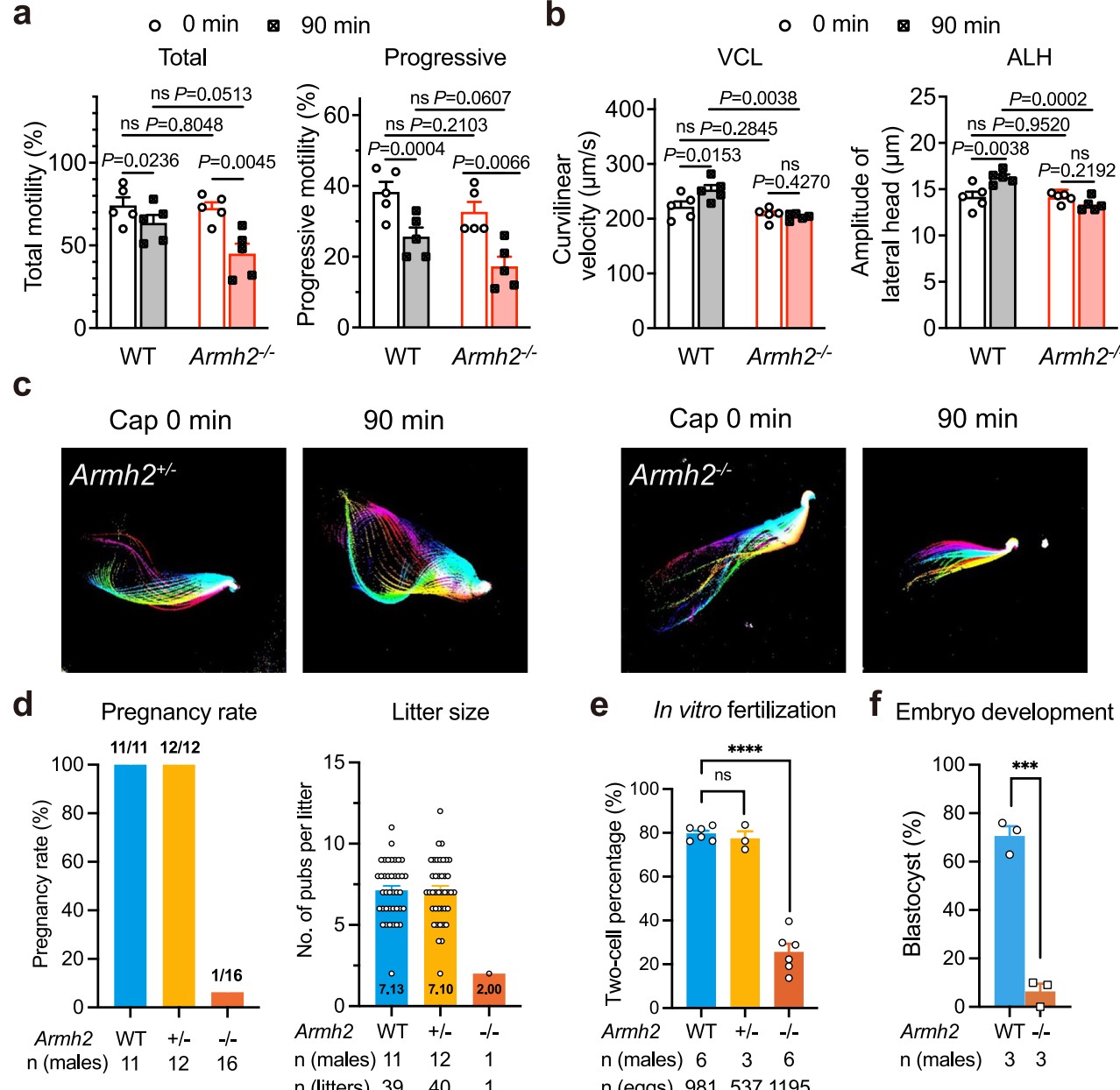

**Fig. 6 | *Armh2*-null male mice are severely subfertile due to defective sperm hyperactivation.** **a**, **b** The motility parameters of WT (black columns) and *Armh2*$^{-/-}$ (red columns) sperm before (hollow dots) and after (filled squares) 90 min incubation in HTF medium, measured by CASA: total and progressive motility (**a**), curvilinear velocity (VCL) and amplitude of lateral head (ALH) (**b**). Data are presented as mean ± SEM. *n* = 5 for each group, where n represents the number of biologically independent male mice. *P*-values are indicated in the figure. ns, no significant difference. **c** Flagellar beating pattern of *Armh2*$^{+/-}$ and *Armh2*$^{-/-}$ spermatozoa incubated under capacitating conditions (Cap) for 0 or 90 min, respectively. Movies were taken at 200 fps. Overlaid colorful images are generated from captured picture stacks in one beat cycle by ImageJ. Data are representative of three independent experiments. **d** Pregnancy rate (left) and litter size (right) of *Armh2*$^{-/-}$ (orange) male mice, compared with WT (blue) and *Armh2*$^{+/-}$ (yellow) males, when mated with WT females over a period of 2–3 months. The litter size data are represented as mean ± SEM. The number of biologically independent male mice and litters for each group is indicated in the figure. The pregnancy rate and average litter size are labeled above and inside the corresponding columns, respectively. **e** In vitro fertilization rates of sperm from WT (blue), *Armh2*$^{+/-}$ (yellow), and *Armh2*$^{-/-}$ (orange) mice incubated with WT oocytes. The percentages of two-cell embryos are presented as mean ± SEM. The number of biologically independent male mice and total oocytes per group is indicated in the figure. Significant difference was observed between *Armh2*$^{-/-}$ and WT group (*P* = 7.45 × 10$^{-6}$). ****P* < 0.0001. ns: no significant difference. **f** Percentage of embryos progressing from the 2-cell stage to blastocyst after in vitro fertilization. Data are presented as mean ± SEM; *n* = 3, where n represents the number of biologically independent male mice. Significant difference was observed between *Armh2*$^{-/-}$ and WT group (*P* = 0.0003). ****P* < 0.001. *P* values were calculated using two-tailed unpaired Student's *t*-tests, except for within-group comparisons in (**a**, **b**), which were assessed using two-tailed paired Student's *t*-tests. Source data are provided in the Source Data file.

suggest that the cytosolic subcomplex forms a functional module in response to intracellular signals for channel gating. Notably, our study implies that the sensitivity of the subcomplex to Ca$^{2+}$ is modulated by intracellular alkalinization (Figs. 5b, e). Compared with the alkaline condition of pH$_i$ = 7.4, the Ca$^{2+}$ sensitivity of $I_{CatSper}$ is reduced at pH$_i$ = 6.0, regardless of NH$_4$Cl treatment (Fig. 5d, e). Although the precise pH$_i$ level after NH$_4$Cl application is unclear, these findings imply that a relatively alkaline intracellular environment is required for

the manifestation of $Ca^{2+}$ sensitivity. This feature supports the notion that intracellular alkalinization serves as the primary physiological signal for CatSper activation.

Interestingly, at a low $pH_i$ of 6.0, we observed similar amplitudes of $I_{CatSper}$ in both $Armh2^{+/-}$ and $Armh2^{-/-}$ sperm (Fig. 5d, e). Given the reduced numbers of CatSper channels in $Armh2^{-/-}$ sperm (Fig. 4c–f), this suggests that CatSper channels lacking the cytosolic subcomplex may exhibit a higher open probability and/or single-channel conductance under low $pH_i$ conditions. In contrast, upon intracellular alkalinization, the fold increase in $I_{CatSper}$ amplitude was larger in $Armh2^{+/-}$ sperm than in $Armh2^{-/-}$ sperm (Fig. 5f), indicating that the cytosolic subcomplex enhances channel activity in response to pH elevation. Therefore, the cytosolic subcomplex may act as a bidirectional regulator: partially suppressing CatSper activity under resting conditions, while promoting robust channel opening upon intracellular alkalinization, a regulatory mechanism critical for precise control of CatSper gating during sperm activation.

### Specific role of ARMH2

According to the structural model, ARMH2, EFCAB9, and CATSPERζ interact with each other through extensive, specific interactions to form a stable subcomplex (Fig. 1, Supplementary Fig. 3). Although it is evident that the subcomplex plays a significant role in regulating CatSper channel gating, the strong interdependence of the three components makes it challenging to specify their individual functions. EFCAB9 has previously been reported to serve as a dual sensor for physiological $Ca^{2+}$ and pH[8]. Whether CATSPERζ and ARMH2 are also directly involved in these processes remains unclear. Notably, the armadillo repeat found in ARMH2 is a structural motif involved in protein-protein interactions that is widely distributed in many proteins with various functions[46]. Based on the structural characterization, we speculate that ARMH2 acts as a scaffold to support and reinforce the cytosolic subcomplex. Additionally, the presence of ARMH2 increases the size of the subcomplex, enabling it to interact with other regions of the channel to better perform the gating function. In support of this notion, we observed additional densities associated with the subcomplex, which may correspond to unresolved segments from the channel components (CATSPER1-4) (Fig. 1c). The close contact between the cytosolic subcomplex and the channel subunits thus provides a structural basis for the transmission of the gating signals sensed by the cytosolic subcomplex to the opening of the CatSper channel.

In summary, we employed an integrated strategy to identify ARMH2 as an evolutionarily conserved cytosolic component of the CatSper channel. We provided evidence that ARMH2 forms a functional module together with the previously characterized EFCAB9 and CATSPERζ, linking pH and $Ca^{2+}$ sensing for CatSper channel gating. Our study offers insights into the regulatory mechanisms of CatSper channel activity and its component organization.

## Methods

### Animals

The animal maintenance and experimental procedures were performed in accordance with institutional guidelines, and all animal studies were approved by the Institutional Animal Care and Use Committee (IACUC) of Westlake University in Hangzhou, China. All mice were maintained on a C57BL/6 background, and those aged 2-5 months were used for experiments in this study. Strict barrier facilities with macroenvironmental temperature (20–26 °C) and humidity (40–70%) were setup. The rooms followed a 12-h light/dark cycle. The investigators were not blinded to the allocation during the experimental procedures or outcome assessments.

### Generation of knockout mice

*Catsper1*, *Efcab9* and *C2cd6* knockout mice were generated from previous studies[8,9,31]. *Armh2* and *Catsperq/Tmem249* knockout mice were generated on a C57BL/6 background using the CRISPR/Cas9 system in this study. A mixture of guide RNA (gRNA1: 5′-TCGCTGCTAT-GACTGGCACC-3′, gRNA2: 5′-TCAGCCTGGCACTAGCCAAC-3′ for *Armh2* knockout; gRNA3: 5′-GTACCATGGAGAATCTCCAC-3′, gRNA4: 5′-CGCCCAGTGGAGATTCTCCA-3′, for *Catsperq/Tmem249* knockout) and Cas9 mRNA was injected into the pronuclei of fertilized eggs. The developing 2-cell embryos were transplanted into pseudo-pregnant females, and the newborn was identified as F0. After checking F0's genotype by PCR amplifying the targeted gene segment and Sanger sequencing, the heterozygotes of F0 were backcrossed with WT C57BL/6 to obtain F1. F2 homozygous knockout mice were obtained by mating F1 mice. The primers for *Armh2* knockout mice genotyping were Forward 1(5′-AACAGCTTCACTCTACCACATGCTC-3′) and Reverse 1(5′-TTTGATAGAGTCAGGATAGCAGCAGC-3′), Forward 2(5′-AACAGCTTCACTCTACCACATGCTC-3′) and Reverse 1(5′-AGGGCAGC-TATTGTATGTCTTTTGC-3′). The primers for genotyping of *Catsperq/Tmem249* knockout mice were Forward 1 (5′-GCTTCTCCGC CCTTGTTAAGTAC-3′) and Reverse 1 (5′- AACGAAGCCCATCT-GAACTCCAG-3′), Forward 2 (5′- TTACCCCTTCAAGCAGCAACAGC-3′) and Reverse 2 (5′- ACGGAAGTGAGTTCTGGCAAGAG-3′).

### Antibodies

The polyclonal antibodies against CATSPER1, CATSPERβ, and ARMH2 were homemade in this study. Peptides corresponding to CATSPER1 (MDQSSRRDESYHETHPGSLDPSHQSHPHPHPHPTLHRPNQGGVYYDSP QHGMFQQPYQQHGGFHQQNELQHLREFSDSHDNAFSHHSYQQDRAGV STLPNNISHAYGGSHPLAESQHSGGPQSGPRIDPNHHPHQDDPHRPSEPL SHPSS) and ARMH2 (KVKRRFFASPQKEEVPTSTADYIFHREKILELGSILKN KRLSLDKRAQAAQKIGLLSFTGGMSAAQFASEYMTEVAFLLQKKKAMSFR TKILLLQSVACWCYLNPDSQRKVR), which were expressed in *E.coli* and purified, and synthetic CATSPERβ (TDNFYHADPSKPIPRN) were conjugated to KLH. The antibodies against CATSPER1 and CATSPERβ were purified from the peptide-immunized rabbits, and the antibody against ARMH2 was purified from the peptide-immunized mice. Rabbit polyclonal antibodies against CATSPERγ, CATSPERδ, CATSPERε, CATSPERθ, CATSPERζ, and EFCAB9 were described previously[14,15,21].

### Western blotting

Cauda sperm pellets of WT or knockout males were resuspended and lysed on ice in buffer A containing 100 mM HEPES-Na, pH 7.5, 1 mM EDTA, 2.5% SDS, 2 mM phenylmethylsulphonyl fluoride (PMSF), 2.6 μg/mL aprotinin, 1.4 μg/mL pepstatin and 10 μg/mL leupeptin. The lysis of whole sperm protein was denatured by boiling at 95 °C for 10 min with 5× SDS loading buffer and cooled down to room temperature for loading into the SDS-PAGE gel. The protein samples were separated by SDS-PAGE and transferred to polyvinylidene difluoride membranes (Biorad, Hercules, USA). The membranes were blocked with 5% non-fat milk (Sangon Biotech, A600669-0250) in TBST solution for 1 h at room temperature and incubated overnight at 4 °C with the following primary antibodies: Anti-mouse CATSPER1 (0.77 μg/mL), CATSPERβ (1.42 μg/mL), CATSPERγ (1 μg/mL), CAT-SPERδ (1.5 μg/mL), CATSPERε (1 μg/mL), CATSPERζ (2.7 μg/mL), CATSPERθ (1.5 μg/mL), EFCAB9 (1 μg/mL), ARMH2 (1:500), and β-actin (1:10,000, Sigma Aldrich). HRP-conjugated goat anti-mouse IgG and goat anti-rabbit IgG were used for secondary antibodies (1:10,000; Jackson ImmunoResearch). The blotting was developed by SuperSignal™ West Pico PLUS Chemiluminescent Substrate (ThermoFisher).

### Cryo-EM sample preparation and data acquisition

The isolated native CatSpermasome sample was prepared as previously reported[16]. For cryo-EM sample preparation, aliquots (3.5 μL) of the protein sample were loaded onto glow-discharged lacey carbon Cu grids coated with ultrathin carbon film (Ted Pella Inc., 400 mesh) or holey carbon grids coated with 2 nm carbon film (Quantifoil Cu

R1.2/1.3 + 2 nm C, 300 mesh). The grids were blotted for 8 s or 3.5 s after waiting for 1 min and immersed in liquid ethane using Vitrobot (Mark IV, Thermo Fisher Scientific) under 100% humidity at 8 °C, respectively. The imaging system comprised of a Titan Krios operating at 300 kV, a Gatan K3 Summit detector, and a GIF Quantum energy filter with a 20-eV slit width. Movie stacks were automatically acquired in super-resolution mode (81,000 × magnification) using EPU software (Thermo Fisher Scientific), with a defocus range from −1.5 μm to −2.5 μm. Each stack was exposed for 2.56 s with 0.08 s per frame, resulting in 32 frames and approximately 50 e⁻/Å² of total dose.

## Cryo-EM data processing and model building

All structure determination procedures were conducted using cryoS-PARC v4[47]. A detailed flowchart outlining the process is provided in Supplementary Fig. 1c. In summary, a total of 34,493 movie stacks were firstly applied for motion correction and patch-CTF estimation. A subset of particles from blob picking were subjected to 2D classification to generate templates for templated-based particle picking. Subsequently, 2D classifications were performed, resulting in approximately 2.7 million particles with clear features. These particles were used for generating an ab initio reconstruction. The particles were then performed for two rounds of heterogeneous refinement, yielding 1,505,415 good particles for non-uniform refinement. Following this, another round of heterogeneous refinement (K = 5) was conducted. 452,960 particles (30%) from the best class, containing the intact complex features, were selected for subsequent non-uniform refinement. To further enhance the resolution of the intracellular portion of the map, particles corresponding to the intracellular components were subtracted, followed by five parallel rounds of local 3D classifications. This process yielded a final set of 83,929 particles classified as high-quality. These particles were further subtracted with a mask around cytosolic map 2 region and subsequently subjected to local refinement, resulting in a final local map with a resolution of 4.8 Å.

The AlphaFold-predicted structure of the ARMH2-EFCAB9-CATSPERζ subcomplex was manually fitted into the density map to serve as the starting model. Subsequent modeling was carried out in Coot[48], ensuring that the main chains conformed to the density map. Structure refinement in real space was carried out in Phenix[49], employing secondary structure and geometry restraints to prevent overfitting. Statistics related to data collection, 3D reconstruction and model refinement can be found in Supplementary Table 1. All Figures in this article related to the ARMH2-EFCAB9-CATSPERζ subcomplex structure were generated using PyMOL[50], Chimera[51], or ChimeraX[52].

## Pull-down assay

Direct interaction between ARMH2 and the CATSPERζ-EFCAB9 subcomplex was evaluated using anti-DYKDDDDK(FLAG) G1 Affinity Resin (GenScript, L00432-25) to pull down the CATSPERζ-EFCAB9 subcomplex with 3×FLAG-tagged ARMH2, or alternatively, using STarm Beads 4FF (Smart-Lifesciences, SA092025) to pull down ARMH2 with the Twin-Strep-tagged CATSPERζ-EFCAB9 subcomplex.

The coding sequences of ARMH2 (49−230), CATSPERζ, and EFCAB9 were individually constructed into the pCAG expression vector, containing a C-terminal 3×FLAG tag, a C-terminal GFP tag, and an N-terminal Twin-Strep tag, respectively. 150 mL HEK293F cells were co-transfected with all three plasmids. As a control, 150 mL HEK293F cells were co-transfected with two plasmids encoding GFP-CATSPERζ and Twin-Strep-EFCAB9. After 60 h post-transfection, cells were collected and cell pellets were resuspended and sonicated in buffer containing 300 mM NaCl, 25 mM HEPES-Na, pH 7.4, protease inhibitors including 2 mM PMSF, 1.3 μg/mL aprotinin, 0.7 μg/mL pepstatin and 0.5 μg/mL leupeptin. After spinning down with 20,000 g at 4 °C for 1 h, the supernatant was separately loaded onto anti-FLAG G1 Affinity Resin and STarm Beads 4FF equilibrated with pre-binding buffer containing 25 mM HEPES-Na, pH 7.0, 400 mM NaCl. The resin was washed with the pre-binding buffer for five times and eluted with 25 mM HEPES-Na, pH 7.4, 150 mM NaCl, added with 250 μg/mL FLAG peptide (for anti-FLAG G1 Affinity resin) or 5 mM D-Biotin (for STarm Beads 4FF). The presence of each component was detected by Western blotting.

## Coevolutionary analysis

The species distribution of the *Armh2* gene was initially assessed in OrthoDB (V.11, orthogroup: 5386992at2759) and supplemented with BLAST searches in the NCBI database. To expand the taxonomic coverage of *Armh2*, a Hidden Markov Model (HMM) was constructed using a representative set of aligned chordate sequences (Supplementary Fig. 3c). The HMM was created with the hmmbuild program from the HMMER suite (version 3.3.2) and used to search 1952 eukaryotic proteomes available in OrthoDB V.11. hmmsearch was employed with an E-value cutoff of $10^{-3}$ to obtain a presence/absence vector for *Armh2* across species.

The resulting presence/absence vector was integrated into the OrthoDB dataset, which included 65,201 orthogroups, and analyzed using the Cotr program[53] to calculate coevolutionary scores and *p*-values (Supplementary Fig. 4). Cytoscape V.3.10.3 was utilized to visualize the network of coevolutionary relationships among CatSper genes. The ggtree v3.1.0 R package[54] was used to visualize CatSper gene presence across eukaryote phylogeny.

A similar HMM-based search was used to identify non-mammalian CATSPERζ sequences, which were missed by BLAST searches, for the comparative reconstruction of the subcomplex in multiple species using AlphaFold 3 (Supplementary Fig. 2b).

## Histology analysis

Testes from WT and *Armh2⁻/⁻* males were fixed in Bouin's fluid (Sigma, HT10132-1L) for 12 - 24 h, washed with PBS, emerged into 75% ethanol for 30 min and dehydrated by HistoCore PEARL-Tissue Processor (Leica). After embedded in paraffin with HistoCore Arcadia Paraffin Embedding Station (Leica), the testes samples were sectioned at 5 μm per slice on HistoCore Microtome (Leica). The paraffin sections were adhered to slides and stained with Hematoxylin & Eosin on HistoCore SPECTRA ST Stainer, and covered by HistoCore SPECTRA Coverslipper (Leica). The slides were imaged with Fluorescence Stereo Zoom Microscope (Zeiss).

## Mating test and in vitro fertilization (IVF)

For the mating test, each male mouse of the genotypes of *Armh2⁻/⁻*, *Armh2⁺/⁻*, and WT male mice was caged with two WT females for 2 - 3 months, respectively. The pregnancy rate of males and litter size were recorded when they gave birth. For IVF, cauda sperm were collected and incubated in a Toyoda Yokoyama Hoshi (TYH) medium (Biorigin) buffer for 90 min at 37 °C under 5% $CO_2$. The capacitated sperm were then added into a TYH drop of fresh oocytes offered by WT super ovulated females. The ratio of spermatozoa and oocyte cell number is 150 - 200:1. 2-cell embryos were counted 24 h after IVF. For developmental analysis, the 2-cell embryos were transferred into potassium simplex optimized medium (KSOM) and monitored over time until reaching the blastocyst stage. Bright-field images were captured at 24-h intervals, and developmental progression was evaluated by cell counting.

## Computer-assisted sperm analysis (CASA)

Spermatozoa were isolated from the cauda epididymis and released into PBS buffer after a 10-min incubation at 37 °C to allow sperm dispersion. The epididymal tissue was carefully removed using forceps, and the sperm-containing media was centrifuged at 200 g for 5 min. The resulting sperm pellet was resuspended in pre-warmed HTF medium (Sigma-Aldrich) and incubated in a tissue culture incubator at 37 °C, 5% $CO_2$ for 10 min (designated as capacitation time of 0 min). Sperm motility was assessed at 0, 15, 30, 60, and 90 min of

capacitation with a computer-assisted semen-analysis system (TOX IVOS II Automatic Sperm Analyzer, Hamilton Thorne Research). Briefly, a 30-μL aliquot of the sperm suspension was loaded onto a 100-μm-depth counting chamber (Leja) and observed under a negative phase-contrast field. Sperm cell tracks were recorded for 0.5 s at 60 Hz. For each specimen and time point, 100–150 spermatozoa were analyzed across ≥5 randomly selected fields to determine total sperm motility (%), progressive sperm motility (%), curvilinear velocity (VCL, μm/s), and amplitude of lateral head displacement (ALH, μm).

### Flagellar bending waveform analysis

Sperm flagellar movement was analyzed as previously described[8]. In short, sperm cells incubated in TYH buffer were transferred to a 35 mm cell culture dish. After settling down for 3 - 5 min, the samples were washed gently to remove unattached cells. Sperm tail movements were recorded for 1 s with a 200 fps speed using Multi-mode Spinning Disk Confocal Microscopy (Olympus, MC-DEMO-001). Beating frequency was measured, and overlaid images were generated by FIJI software as previous study[15,55].

### Quantitative proteomic analysis

To build up an in-house mouse sperm mass spectral library for quantitative mass spectrometry (MS) analysis, a mouse sperm pooled sample was initially lysed with 8 M urea and digested using pressure cycling technology (PCT) assisted tryptic digestion[56]. The peptides were then separated into 20 fractions over a 60-min gradient using basic pH reverse-phase liquid chromatography (LC) (Dionex Ultimate 3000 UPHLC, Thermo Fisher Scientific). The fractioned peptides were injected into a Q Exactive HF mass spectrometer (QE-HF MS, Thermo Fisher Scientific) using data-dependent acquisition (DDA) mode. The DDA-MS raw files of the mouse sperm samples were searched using FragPipe (version 18.0), integrating with MSFragger[57] (version 3.4), against the Mus musculus FASTA from UniProtKB containing 63,596 mouse proteins (downloaded on April 18, 2023) to build an in-house library for data-independent acquisition (DIA) analysis.

Cauda sperm samples from WT and $Armh2^{-/-}$ mice ($n = 4$ per group) were then collected and lysed with urea in triplicate for quantitative DIA MS measurement. The lysates were separated by SDS-PAGE, and peptides were prepared as previously described[43]. Briefly, for data-dependent acquisition (DDA) identification, each lane was cut into ten bands, and each gel band was cut into 1 mm³ pieces. The resulting gel pieces were reduced with 10 mM Tris (2-carboxyethyl) phosphine hydrochloride (TCEP) in 25 mM $NH_4HCO_3$ for 1 h at 25 °C, followed by alkylation with 55 mM iodoacetamide in 25 mM $NH_4HCO_3$ solution in the dark for 30 min at 25 °C. Sequential tryptic digestion was then performed at an enzyme concentration of 12.5 ng/μL at 37 °C overnight. The tryptic-digested peptides were extracted with 1% formic acid in a 50% acetonitrile aqueous solution, repeated twice. The solution was then dried under vacuum and further cleaned by Pierce C18 Spin Tips (Thermo Fisher Scientific). 500 ng peptides were injected into QE-HF MS (Thermo Fisher Scientific) for DIA MS analysis.

MS settings are listed as follows. Briefly, the sample was firstly separated at 300 nL/min in a 60-min of 3–28% linear liquid chromatography gradient (buffer A: 2% acetonitrile, 0.1% formic acid; buffer B: 98% acetonitrile, 0.1% formic acid). Eluting peptides were ionized at a potential of +1.8 kV in QE-HF MS. For DDA mode, the full MS scan was acquired by analyzing the 400–1200 m/z range at a 60,000-resolution in the Orbitrap using a maximum injection time of 80 ms. After the full MS scan, the top 20 precursors were fragmented in the 200–2000 m/z range at a 30,000-resolution using a maximum injection time of 100 ms. For DIA mode, the full MS scan was acquired by analyzing the 390–1010 m/z range at a 60,000-resolution in the Orbitrap using a maximum injection time of 80 ms. After the full MS scan, 24 MS/MS scans were acquired, each with a 30,000-resolution, a normalized collision energy of 28%, and the default charge state set to 2, with the

maximum injection time set to auto. The cycle of 24 MS/MS scans with the wide isolation window was as follows (m/z): 410, 430, 450, 470, 490, 510, 530, 550, 570, 590, 610, 630, 650, 670, 690, 710, 730, 770, 790, 820, 860, 910, 970.

The DIA MS data were analyzed by DIA-NN[58] (version 1.8.1) using the in-house library (6746 proteins) following parameters: a 1% false discovery rate (FDR) for precursors, fixed modifications including "C carbamidomethylation" and variable modifications including "N-terminal methionine excision (NME)". Our approach involved selecting "Unrelated runs" and "Use isotopologues". The quantification strategy was set to "Robust LC (high precision)". Other settings were left as default values. The comparison between the knockout and WT group was performed using a two-sided unpaired Welch's t-test. Differentially expressed proteins were identified based on a combination of $P < 0.05$ and an absolute $\log_2$ (Fold change) $> 0.5$.

### Generation of $Armh2^{-/-}$; $Catsper1^{GFP/GFP}$ mice

The $Catsper1^{GFP/GFP}$ mice were generated in our previous study by knocking in a fragment encoding 3×FLAG-EGFP-TEV protease cutting site at the N-terminus of the $Catsper1$ gene locus[16]. The $Catsper1^{GFP/GFP}$ male mice were mated with $Armh2^{-/-}$ female mice to generate double heterozygotes. Double homozygotes ($Armh2^{-/-}$; $Catsper1^{GFP/GFP}$) were then acquired by mating between the offsprings of double heterozygotes. The genotyping of the mice was confirmed by PCR.

### Super-resolution imaging

Sperm from $Catsper1^{GFP/GFP}$ and $Armh2^{-/-}$; $Catsper1^{GFP/GFP}$ mice were used for a super-resolution imaging study. Fresh sperm were washed twice in PBS, attached to the poly-l-lysine-coated (Sigma-Aldrich, P4707) 25 mm coverslip (CG15XH, Thorlabs) for 10 min, and fixed with 4% paraformaldehyde in PBS at room temperature for 15 min. Unbound sperm cells were removed by rinsing three times with PBS. Fixed sperm cells were treated with permeabilization and blocking buffer containing 0.2% TritonX-100, 3% BSA in PBS at room temperature for 1 h. The anti-GFP antibody (1:500; Thermo Fisher, A-11122) was diluted with 1% BSA, 0.05% TritonX-100 in PBS, and incubated with sperm cells at 4 °C overnight. The samples were then incubated with goat anti-rabbit IgG (H + L) Alexa Fluor™ 647 (1:1000; Thermo Fisher, A21245) in dilution buffer at room temperature for 1 h. Following antibody labeling, the samples were washed three times with 0.1% TritonX-100 in PBS for 5 min each. Lastly, the samples were rinsed three times in PBS.

The imaging buffer for super-resolution imaging was prepared every time immediately before use, where catalase and glucose oxidase were diluted in base buffer containing 44% glycerol, 50 mM Tris, pH 8.0, 10 mM NaCl, 10% glucose, and 35 mM cysteamine hydrochloride. Super-resolution imaging was performed on a custom-built 4Pi single-molecule switching (4Pi-SMS) microscope as previously described with minor modifications[59,60]. Briefly, the emitted fluorescent signals were collected coherently through a pair of opposing objectives (Olympus, 100×/1.35 NA, silicone oil immersion) and imaged with an sCMOS camera (Hamamatsu, ORCA-Fusion BT). The microscope was equipped with an excitation laser at 642 nm (MPB Communications, 2RU-VFL-2000-642-B1R) and an activation laser at 405 nm (Coherent OBIS 405 LX, 100 mW). All data were acquired at 100 fps at a 642 nm laser intensity of approximately 8 kW/cm². The 4Pi-SMS images and movies were rendered using Vutara SRX software (Bruker).

### Electrophysiology

The whole-cell patch-clamp recordings of mouse sperm were performed as previously described[5,8,39,40]. Briefly, Corpus sperm cells were freed and swam into a HEPES-buffered saline (HS) buffer containing (mM): 135 NaCl, 5 KCl, 1 $MgSO_4$, 2 $CaCl_2$, 20 HEPES, 5 glucose, 10 lactic acid, and 1 Na pyruvate. Buffer pH was adjusted to 7.40 with NaOH, and osmolarity was adjusted to ~300 mOsm/L with glucose. The sperm

head were then allowed to settle down and attach to a 35-mm cell culture dish. Micropipettes were pulled with a P-1000 flaming/Brown Micropipette Puller System (Sutter Instrument) and fire-polished with Micro Forge MF2 (Narishige) from the 1.5/1.2 mm (outer diameter/inner diameter), thin-walled glass (Sutter Instrument). The series resistance of micropipettes was typically 15–20 MΩ. The pipette internal solution recipe for CatSper channel recording contained (in mM): 5 CsCl, 10 EGTA, 20 HEPES. Solution pH was adjusted to 7.40 with CsOH, and osmolarity was adjusted to ~300 mOsm/L with glucose. After the formation of gigaohm seals at the cytoplasmic droplet, voltage pulses from 350 mV to 650 mV and simultaneous suction were applied to break in the cell membrane. Whole cell currents were then recorded in a divalent-free (DVF) solution containing (in mM): 150 CsMeSO$_3$, 5 EDTA, 20 HEPES. Solution pH was adjusted to 7.40 with CsOH, and osmolarity was adjusted to ~300 mOsm/L with glucose. To evaluate the Ca$^{2+}$ sensitivity of CatSper, the following pipette solutions were used. For pH 6.0 with 0 or 0.1 μM free Ca$^{2+}$, the solution contained (in mM): 135 mM CsMeSO$_3$, 5 or 4.63 CsCl, 0 or 0.185 CaCl$_2$, 1 EGTA, 1 EDTA, 1 BAPTA, and 20 MES. For pH 6.0 with 10 μM free Ca$^{2+}$, the solution contained (in mM): 135 mM CsMeSO$_3$, 4.54 CsCl, 0.23 CaCl$_2$, 3 HEDTA, and 20 MES. For pH 7.4 with 0 or 0.1 μM free Ca$^{2+}$, the solution contained (in mM): 135 mM CsMeSO$_3$, 5 or 1.52 CsCl, 0 or 1.74 CaCl$_2$, 1 EGTA, 1 EDTA, 1 BAPTA, and 20 HEPES. For pH 7.4 with 10 μM free Ca$^{2+}$, the solution contained (in mM): 135 mM CsMeSO$_3$, 0.67 CsCl, 2.165 CaCl$_2$, 3 HEDTA, and 20 HEPES. Desired free Ca$^{2+}$ concentrations were calculated by WINMAXC32 version 2.51 (Chris Patton, Stanford University). For intracellular alkalization activation (pH$_i$ = ~7.4), 10 mM NH$_4$Cl were added to the bath solution. The holding potential was 0 mV. Ramp curves were applied by a continues slope voltage stimulus from −100 to +100 mV. IV curves were generated using a group of step potentials ranging from −100 to +100 mV, with a 20 mV increment. Data were performed with EPC10 amplifier with PatchMaster software (HEKA) at room temperature (23 ± 2 °C), sampled at 25 kHz and filtered at 1 kHz, and analyzed with Mini Analysis software (BlueCell, v6.0.3). All Figures were plotted with Graphpad Prism software (Dotmatics, v10.0.2).

### Reporting summary

Further information on research design is available in the Nature Portfolio Reporting Summary linked to this article.

## Data availability

The cryo-EM map and atomic coordinate of the ARMH2-EFCAB9-CATSPERζ subcomplex have been deposited at the Electron Microscopy Data Bank (EMDB) and the Protein Data Bank (PDB) under the accession codes of EMD-63452 and 9LWO, respectively. The proteomics raw data have been deposited in iProX under the accession code of IPX0013365000. The source data underlying Figs. 2a, 2e, 2f, 3, 4a-e, 5b, 5e, f, 6a, b, 6d–f and Supplementary Fig. 5b–d, 6a–c, 7b, 8a are provided as a Source data file.

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

## Acknowledgments

We thank the Cryo-EM Facility and High-Performance Computing Center of Westlake University for providing cryo-EM and computation support; X. Ge, D. Wu, Y. Yang and the Laboratory Animal Resources Center of Westlake University for help with animal maintenance and IVF experiments; and G. Fang and the Imaging Core of Westlake University for help with imaging training and assistance. This work was supported by the National Natural Science Foundation of China (32271261 to J.W., 32271239 to Z.Y., 32271167 and 82471646 to X.Z., 323B2037 to S.L., and 82401881 to H.K.), Zhejiang Provincial Natural Science Foundation of China (LR22C050003 and LDG25C050002 to J.W.), Key R&D Program of Zhejiang (2024SSYS0029), Westlake University (1011103860222B1 to J.W. and 1011103560222B1 to Z.Y.) and Westlake Education Foundation (101486021901 to J.W. and 101456021901 to Z.Y.).

## Author contributions

J.W. conceived the project. J.W., Z. Yan, X.Z., J.-J.C., R.P., M.J., Y. Zhang, M.L., and Y. Zhu designed and supervised the experiments. Q.Z., S.L., Y.R., X.H., S.W., and S.S. performed most biochemical and functional experiments, including pull-down assay, western blotting, sperm motility and fertility analysis. S.L. prepared the cryo-EM sample and collected the cryo-EM data. S.L. and Q.X. processed the cryo-EM dataset, built the model, and performed structural analysis. H.K. and Q.Z. performed electrophysiological experiments. C.R. and G.S. performed bioinformatics analysis. R.S. and H.C. performed mass spectrometry experiments. Z. Yu and Q.Z. performed super-resolution imaging experiments. All authors analyzed the results. J.W., Z. Yan, and Q.Z. wrote the manuscript with input from all co-authors.

## Competing interests

The authors declare no competing interests.

## Additional information

[1]Fudan University, Shanghai, China. [2]Key Laboratory of Structural Biology of Zhejiang Province, State Key Laboratory of Gene Expression, School of Life Sciences, Westlake University, Hangzhou, Zhejiang, China. [3]Westlake Laboratory of Life Sciences and Biomedicine, Hangzhou, Zhejiang, China. [4]Institute of Biology, Westlake Institute for Advanced Study, Hangzhou, Zhejiang, China. [5]Institute of Reproductive Medicine, Medical school, Nantong University, Nantong, Jiangsu, China. [6]Key Laboratory of Growth Regulation and Translational Research of Zhejiang Province, School of Life Sciences, Westlake University, Hangzhou, Zhejiang, China. [7]School of Life Sciences, Westlake University, Hangzhou, Zhejiang, China. [8]Department of Cellular and Molecular Physiology, Yale School of Medicine, Yale University, New Haven, CT, USA. [9]Department of Chemistry, Life Sciences, and Environmental Sustainability, University of Parma, Parma, Italy. [10]State Key Laboratory of Reproductive Medicine and Offspring Health, Department of Histology and Embryology, School of Basic Medical Sciences, Nanjing Medical University, Nanjing, China. [11]School of Medicine, Westlake University, Hangzhou, Zhejiang, China. [12]Westlake Center for Intelligent Proteomics, Westlake Laboratory of Life Sciences and Biomedicine, Hangzhou, Zhejiang, China. [13]These authors contributed equally: Qingqing Zhao, Shiyi Lin, Hang Kang. ✉e-mail: zengxuhui@ntu.edu.cn; yanzhen@westlake.edu.cn; wujianping@westlake.edu.cn

