## [Transparent Peer review file · Nature Communications]

ARMH2 is a cytosolic component of CatSper crucial for sperm function

Corresponding Author: Professor Jianping Wu

Version 0:

Reviewer comments:

Reviewer #1

(Remarks to the Author)

The manuscript by Zhao et al., reports the expression of ARMH2, a novel uncharacterized cytosolic component of the CatSper complex. Briefly, sperm capacitation and fertilization are highly regulated processes which depend on external Ca²⁺, where the sperm-specific Ca²⁺ channel, CatSper, plays a key role in sperm hyperactivated motility and fertility regulating the agonist-induced Ca²⁺ influx. CatSper channel is a transmembrane protein complex, constituted by four pore-forming CATSPER (1-4) and nine auxiliary subunits (β - ; plus SLCO6C1, CATSPER η (TMEM262), CATSPER θ (TMEM249), and the cytosolic proteins CATSPER ζ and EFCAB9). However, previous studies of the mouse CatSper channel suggested the presence of remaining unidentified component(s).

In the present manuscript, authors identify the protein armadillo-like helical domain containing 2 (ARMH2) as a novel component of CatSper, by combining cryo-EM, mass spectrometry, AlphaFold structure prediction, and coevolutionary analysis. ARMH2 forms a cytosolic ternary subcomplex with EFCAB9 and CATSPER ζ , which contributes to the stable assembly of the CatSper nanodomains along the sperm tail and regulates its pH and Ca²⁺ sensitivity. Loss of ARMH2 leads to compromised physiological activation of CatSper, thereby resulting in asthenozoospermia and severe subfertility.

What are the noteworthy results?

The results shown in this manuscript indicate that ARMH2 is crucial for sperm function and provide new insights into the composition and functional regulation of CatSper.

Will the work be of significance to the field and related fields? How does it compare to the established literature? If the work is not original, please provide relevant references.

Authors have contributed to the structural and functional characterization of CatSper channel in previous reports. The topic presented in this manuscript is relevant for the field, showing that the CatSper structure is more complex than though.

Does the work support the conclusions and claims, or is additional evidence needed?

Yes, the results shown in this manuscript completely support conclusions and claims

Are there any flaws in the data analysis, interpretation and conclusions? Do these prohibit publication or require revision?

The interpretations and conclusions are well supported by data analysis. I recommend to publish the present manuscript. Is the methodology sound? Does the work meet the expected standards in your field?

As above mentioned, authors have contributed to the structural and functional characterization of CatSper channel in previous reports. In the present manuscript they combined cryo-EM, mass spectrometry, AlphaFold structure prediction, and coevolutionary analysis to identify a new component of the CatSper complex.

Is there enough detail provided in the methods for the work to be reproduced?

Yes, indeed.

Authors have contributed to the structural and functional characterization of CatSper channel in previous reports. The topic presented in this manuscript is relevant for the field, showing that the CatSper structure is more complex than though. The manuscript is well written, and results are well presented.

Major comments:

None

Minor comments:

1.- Page 2, line 33. Please, substitute the phrase: "Sperm capacitation and fertilization is highly regulated by Ca²⁺ signaling" by "Sperm capacitation and fertilization are highly regulated by Ca²⁺ signaling".

2.- Please check CatSperq. Page 8, line 184. This gene hasn't been described, at least not yet.

Reviewer #2

(Remarks to the Author)

Cytoplasmic Ca^{2+} ($[\text{Ca}^{2+}]_i$) is a key regulator of the main sperm functions. The principal Ca^{2+} channel of sperm is CatSper, which is exclusive of this cell. It is essential for hyperactivated motility and fertilization. So far CatSper is the most complex channel known, constituted by 4 pore-forming subunits (CATSPER1-4) and multiple auxiliary subunits. Previous studies regarding this channel had left undefined remaining component(s). The authors combine cryo-EM, mass spectrometry, AlphaFold structure prediction, and coevolutionary analysis, to identify a novel CatSper component, armadillo-like helical domain containing 2 (ARMH2). This component is assembled in a ternary subcomplex with EFCAB9 and CATSPER ζ in the cytosol that is involved in stabilizing a CatSper nanoscopic linear arrangement along the sperm tail which regulates its pH and Ca^{2+} sensitivity. The lack of ARMH2 diminishes CatSper's physiological activation and produces asthenozoospermia and severe subfertility. The observations presented by the authors indicate that that ARMH2 significantly influences sperm function and reveal structural details of CatSper that contribute to explain its behavior. The use of complementary novel methodologies to identify ARMH2 offers new strategies to unravel protein complex structure-function relationships.

This work presents a significant advance in the structure-function relationship of the CatSper channel, the main sperm Ca^{2+} channel that is key for motility and fertilization. As this is the most complex Ca^{2+} channel known, the findings presented impact Cell Biology in general. The two groups that participate in this work have made important contributions to the field. In general, the work is well presented with solid results that support the author's conclusions. There are however some matters that should be resolved before the work is published that I will state below.

- 1) There seems to be a problem regarding a misfit between text and Extended Data Fig. 2d. There is no such fig. in my Extended fig. file. There is no fig. representing the described results. The confusion is that it is fig. 2d not Extended fig.2d.
- 2) The bright field images in Fig. 5f are very dim lacking contrast, it would be good to improve them.
- 3) In the legend of Extended Fig. 6 sperm is repeated.
- 4) Why are the currents reduced at positive potentials as $[\text{Ca}^{2+}]_i$ is increased? It would be good to determine if the reversal potential changes as $[\text{Ca}^{2+}]_i$ is increased. This is also the case also for the measurements presented as pH_i changes.
- 5) How do the swimming parameters vary as a function of capacitation time in control vs Armh2 null? It seems 90 min is a fairly long time.
- 6) It is unclear to me how the results in Fig. 6A show that hyperactivation is reduced in Armh2 $^{-/-}$ sperm. I can not see that wild sperm hyperactivate, as the percentage of progressive motility does not change with 90 min capacitation. In contrast, the null sperm seem to hyperactivate, as their progressive motility is reduced. This contradicts Fig. 6b, where it is clearly seen that Armh2 $^{-/-}$ sperm do not hyperactivate but wild ones do??
The authors should show the values of hyperactivation.
- 7) The details in Methods of the CASA experiments are really superficial. Why not measure directly hyperactivation in wild and mutant sperm. There are problems with these results.

Reviewer #3

(Remarks to the Author)

In this manuscript by Zhao Q. et al the authors reveal that ARMH2 is a novel component of the CatSpermasome. The authors perform new cryo-EM analysis (building upon a previous publication) to improve the resolution of the cytosolic regions of CatSper, resulting in an overall map in the cytosolic region at $\sim 5\text{\AA}$ resolution. The placement of ARMH2 within this newly resolved map is supported by extensive AlphaFold predictions, pulldown assays, and co-evolution analysis. Further experiments with CRISPR knockout mouse models and electrophysiology demonstrate that ARMH2 is a critical intracellular component of the CatSpermasome.

I approached this review focusing on the electron microscopy aspects of the manuscript, and will leave comments on the other sections (mouse models, electrophysiology, sperm motility) to experts in these particular areas. Although the obtained resolution in cytosolic map 2 is limited to $\sim 5\text{\AA}$, the clearly resolved secondary structure elements strongly support docking of an AlphaFold model of ARMH2. This is a great example of how even intermediate resolution cryo-EM maps from disordered and poorly resolved regions can be integrated with AlphaFold structure prediction and co-evolution analysis to gain important biological insights.

The electron microscopy experiments appear to be well performed, and support the primary conclusions of the manuscript. Overall, I am very supportive of publication of this manuscript. I have only relatively minor comments listed below...

1. In the first paragraph of the results section, it is stated "To facilitate reliable component assignment in this region, we further optimized the cryo-EM sample". However, the methods section only states that "The isolated native CatSpermasome sample were prepared as previously reported", and the vitrification conditions appear to be the same as the previous publication. It is unclear to me what was actually optimized in the current manuscript? Were there any changes to the sample purification/vitrification protocol, or was it simply just a larger dataset? More details on exactly how the sample was "optimized" are needed.
2. For particle picking, the methods and Extended Data Figure 1 only mention "iterative particle picking". More details on

how this picking was performed are warranted. Was this blob picking followed by template-based picking in CryoSparc? Or was a more advanced particle-picking scheme, such as one of the neural-net based pickers employed?

3. From Extended Data Figure 1 and the methods section it is not clear what data processing steps were performed in Relion versus CryoSPARC. I presume the iterative local classification was performed in Relion? But what about the signal subtraction step prior to classification, was this performed in CryoSPARC before exporting the subtracted particles for classification in Relion? Similarly, it is not clear whether or not the final focused refinement to obtain the 4.8Å map was performed in Relion or CryoSPARC. Please expand the methods section and/or Extended Data Figure 1 to include these details.

4. Was 3D classification/refinement in Relion 5.0 performed with or without BLUSH regularization? If it was performed without, have the authors considered trying BLUSH regularization both in the local 3D classification and final refinement steps? For some targets we have found BLUSH regularization to provide significantly improved resolutions and map interpretability, particularly on challenging small proteins.

5. Was the final refinement to obtain the improved resolution of cytosolic map-2 performed on the signal subtracted particles, or was the final refinement performed with the complete (non-subtracted) particles with just a mask around the cytosolic region?

Minor:

1. In Figure 1D it is a little hard to see the atomic model in the transparent volume. Maybe consider increasing the map transparency slightly, particularly for the ARMH2 subunit.

2. Second to last sentence in the second paragraph of the results section states "The cryo-EM resolved regions of the ARMH2-EFCAB9-CATSPERζ subcomplex are primarily in areas with high predicted local distance difference test (pLDDT) scores, supporting the assertion that the unassigned density belongs to ARMH2 (Fig. 1B)". I believe this should point to Fig. 1D instead?

3. "Alternatively, we transiently co-expressed ARMH2-3xFlag, Strep-EFCAB9, and CATSPERζ-GFP in HEK293 cells and examined whether purification of one component would pull-down the other two proteins (Extended Data Fig. 2d)." I believe this should point to Main Figure 2D, not Extended Data Fig. 2D.

4. "Immunoblotting analysis of the ARMH2-null sperm revealed that the detected transmembrane subunits, including CATSPER1, CATSPERβ, δ, and θ, were all reduced to approximately half or less compared to WT, whereas CATSPERζ was entirely absent (Fig. 4a and Extended Data Fig. 4b). There is no western blot or pulldown data in Extended Data Fig. 4b

5. "The cytosolic ternary subcomplex is likely involved in these intercellular junctions" – replace intercellular with intracellular

6. Last word of Fig. 1 legend. "labelled" should be "labeled"

7. First sentence of Fig. 2a legend. "Predicated" should be predicted

Reviewer #4

(Remarks to the Author)

This manuscript by Zhao et al. is a logical follow-up study to this group previous manuscript describing CatSper structure. In that work, the authors used a combination of cryo-EM and mass spectrometry of what they called the CatSpermasome. In addition to all CatSper subunits, several other proteins were found including Armh2. In the present manuscript, further evaluation of cryo-EM coupled to the analysis of predicted structures using AlphaFold 2 predicted ARMH2 to be part of a ternary complex with EFCAB9 and CATSPERζ. Consistent with this prediction: phylogenetic analysis showed co-evolution of Armh2 with other CatSper subunits and null mutant mice are severely subfertile. Moreover, sperm from the Armh2 KO are phenotypically similar to those of Efcab9 and Catsperz null mice. Interestingly, absence of Armh2 reduced the overall levels of the pore-forming CatSper channel subunits, disrupted CatSper structure and eliminated pH dependence of CatSper currents. Overall, this work is excellent and used a very complete set of state-of-the-art methodologies to study the CatSper complex in general and Armh2 in particular. I have only a few comments.

Specific points

- In mouse, Armh2 is known as 17000 16Grik. ARMH2 in Humans is also known as C6orf229. Armh2 is not well represented in mouse sperm proteomic databases. However, it can be found with the alternative name (Grahn et al. DOI: 10.1038/s41467-023-40855-0). This information should be included.

- I was not able to find the quantitative proteomic table used for the generation of Fig. 4c. Please include an excel table as supplementary information comparing results from wild type and Armh2^{-/-} sperm proteomes. This table will be a very important resource for the field.

- In vitro fertilization measured by the percentage of two-cell is about 30 %. Do the authors know the percentage of these 2-cells that reached the blastocyst stage?

Version 1:

Reviewer comments:

Reviewer #1

(Remarks to the Author)

In my opinion, authors have addressed all reviewer's questions in a proper way and they have also included complementary assays to reinforce their results like new initial conditions for computer assisted sperm cell analysis experiments (CASA), Additional developmental analysis for the in vitro fertilization (IVF) experiments, or the addition of western and proteomic data. The results shown in this revised manuscript indicate that ARMH2 is crucial for sperm function and they provide new insights into the composition and functional regulation of CatSper.

Reviewer #2

(Remarks to the Author)

The authors have properly dealt with the majority of the suggestions and questions posed by the reviewers. There are a couple of Figs., like Supp. Fig. 8a where the Y axes should say Curvilinear and it says Curve linear. On line 322 of the revised manuscript the authors have repeated analysis, please correct. I believe the paper is ready to be published.

Reviewer #3

(Remarks to the Author)

The authors have properly addressed all of my critiques. I am supportive of publishing this work without further revision.

Reviewer #4

(Remarks to the Author)

Thee authors have addressed all my queries.

Response to Referees' Comments (NCOMMS-25-17935):

We are very grateful for the referees' thoughtful evaluation of our work and their insightful suggestions, which have helped us strengthen the manuscript. Prior to responding to the referees' individual comments, we provide an overview of the major modifications incorporated into the revised manuscript.

1) Repeat of CASA experiment. Per referee#2's suggestion, we recognized that the initial condition for computer assisted sperm cell analysis (CASA) experiments were suboptimal, probably due to the use of homemade BWW buffer. We have replaced the capacitation buffer to commercially available HTF medium and repeated the CASA experiments. The hyperactivated motility parameters of the WT sperm are now normal, as evidenced by the increased of VCL and ALH, suggesting that the updated CASA results are reliable. These revised data have been incorporated into the revised manuscript (**Fig. 6a,b and Supplementary Fig. 8a**).

2) Addition of developmental analysis for the IVF experiment. To further investigate the percentage of 2-cell embryos that reach the blastocyst stage in the *in vitro* fertilization (IVF) experiments, we performed additional IVF experiments and extended *in vitro* embryo culture to the blastocyst stage (**Fig. 6f and Supplementary Fig. 8b, c**). The new data indicate that the actual IVF successful fertilization rate of *Armh2*^{-/-} sperm is much lower than 20%, as previously estimated based on the the percentage of 2-cells, which aligns more closely with the *in vivo* results.

3) Addition of western blot data. We performed western blot for *Armh2*^{-/-} sperm with more antibodies (anti-EFCAB9, CATSPER γ and ϵ) (**Fig. 4c**). In addition, we also performed anti-ARMH2 western blot in the sperm from various CatSper knockout mice including *Catsper1*^{-/-}, *Efcab9*^{-/-} and *C2cd6*^{-/-} (**Fig. 4a**). The new data consistently support the interdependency between ARMH2 and other CatSper components.

4) Addition of proteomic data. We provided the quantitative proteomic table (excel) as **Supplementary Data 1** in the revised manuscript.

The revised manuscript now contains **6** main figures, **8** supplementary figures, **1** supplementary table, **2** supplementary videos and **1** supplementary Data.

Below are our point-to-point responses (**in blue**) to referees' comments.

Referee #1:

The manuscript by Zhao et al., reports the expression of ARMH2, a novel uncharacterized cytosolic component of the CatSper complex. Briefly, sperm capacitation and fertilization are highly regulated processes which depend on external Ca²⁺, where the sperm-specific Ca²⁺ channel, CatSper, plays a key role in sperm hyperactivated motility and fertility regulating the agonist-induced Ca²⁺ influx. CatSper channel is a transmembrane protein complex, constituted by four pore-forming CATSPER (1-4) and nine auxiliary subunits (β -e; plus SLCO6C1, CATSPER η (TMEM262), CATSPER θ (TMEM249), and the cytosolic proteins CATSPER ζ and EFCAB9). However, previous studies of the mouse CatSper channel suggested the presence of remaining unidentified component(s).

In the present manuscript, authors identify the protein armadillo-like helical domain containing 2 (ARMH2) as a novel component of CatSper, by combining cryo-EM, mass spectrometry, AlphaFold structure prediction, and coevolutionary analysis. ARMH2 forms a cytosolic ternary subcomplex with EFCAB9 and CATSPER ζ , which contributes to the stable assembly of the CatSper nanodomains along the sperm tail and regulates its pH and Ca²⁺ sensitivity. Loss of ARMH2 leads to compromised physiological activation of CatSper, thereby resulting in asthenozoospermia and severe subfertility.

What are the noteworthy results?

The results shown in this manuscript indicate that ARMH2 is crucial for sperm function and provide new insights into the composition and functional regulation of CatSper.

Will the work be of significance to the field and related fields? How does it compare to the established literature? If the work is not original, please provide relevant references.

Authors have contributed to the structural and functional characterization of CatSper channel in previous reports. The topic presented in this manuscript is relevant for the field, showing that the CatSper structure is more complex than thought.

Does the work support the conclusions and claims, or is additional evidence needed?

Yes, the results shown in this manuscript completely support conclusions and claims

Are there any flaws in the data analysis, interpretation and conclusions? Do these prohibit publication or require revision?

The interpretations and conclusions are well supported by data analysis. I recommend to publish the present manuscript

Is the methodology sound? Does the work meet the expected standards in your field?

As above mentioned, authors have contributed to the structural and functional characterization of CatSper channel in previous reports. In the present manuscript they combined cryo-EM, mass spectrometry, AlphaFold structure prediction, and coevolutionary analysis to identify a new component of the CatSper complex.

Is there enough detail provided in the methods for the work to be reproduced?

Yes, indeed.

Authors have contributed to the structural and functional characterization of CatSper

channel in previous reports. The topic presented in this manuscript is relevant for the field, showing that the CatSper structure is more complex than thought. The manuscript is well written, and results are well presented.

We sincerely appreciate the referee's time and effort in reviewing our manuscript. Below, we provide a detailed point-by-point response to the referee's comments, along with the corresponding revisions made to the text.

Major comments:

None

Minor comments:

1.- Page 2, line 33. Please, substitute the phrase: "Sperm capacitation and fertilization is highly regulated by Ca²⁺ signaling" by "Sperm capacitation and fertilization are highly regulated by Ca²⁺ signaling".

We thank the reviewer for the helpful suggestion. We have revised the phrase as recommended.

2.- Please check CatSperq. Page 8, line 184. This gene hasn't been described, at least not yet.

We apologize for the confusion and have revised the sentence to make it clearer. The sentence has been revised to: The extent of Armh2 knockout effect differs from the CatSperq (encoding the transmembrane component CATSPERθ) knockout sperm, in which case all CatSper components are undetectable (Fig. 4c).

We thank this referee for their critical reading and constructive comments.

Referee #2:

Cytoplasmic Ca^{2+} ($[\text{Ca}^{2+}]_i$) is a key regulator of the main sperm functions. The principal Ca^{2+} channel of sperm is CatSper, which is exclusive of this cell. It is essential for hyperactivated motility and fertilization. So far CatSper is the most complex channel known, constituted by 4 pore-forming subunits (CATSPER1-4) and multiple auxiliary subunits. Previous studies regarding this channel had left undefined remaining component(s). The authors combine cryo-EM, mass spectrometry, AlphaFold structure prediction, and coevolutionary analysis, to identify a novel CatSper component, armadillo-like helical domain containing 2 (ARMH2). This component is assembled in a ternary subcomplex with EFCAB9 and CATSPER ζ in the cytosol that is involved in stabilizing a CatSper nanoscopic linear arrangement along the sperm tail which regulates its pH and Ca^{2+} sensitivity. The lack of ARMH2 diminishes CatSper's physiological activation and produces asthenozoospermia and severe subfertility. The observations presented by the authors indicate that that ARMH2 significantly influences sperm function and reveal structural details of CatSper that contribute to explain its behavior. The use of complementary novel methodologies to identify ARMH2 offers new strategies to unravel protein complex structure-function relationships.

This work presents a significant advance in the structure-function relationship of the CatSper channel, the main sperm Ca^{2+} channel that is key for motility and fertilization. As this is the most complex Ca^{2+} channel known, the findings presented impact Cell Biology in general. The two groups that participate in this work have made important contributions to the field.

In general, the work is well presented with solid results that support the author's conclusions. There are however some matters that should be resolved before the work is published that I will state below.

We are grateful to this referee for acknowledging our work and offering kind suggestions. We have carefully addressed the comments and incorporated additional data where necessary to further strengthen the manuscript. Below is our detailed, point-by-point response.

1) There seems to be a problem regarding a misfit between text and Extended Data Fig. 2d. There is no such fig. in my Extended fig. file. There is no fig. representing the described results. The confusion is that it is fig. 2d not Extended fig.2d.

Thank you very much for your kind comment. We apologize for the confusion. "Extended Data Fig. 2d" should be and has been replaced with "Fig. 2d".

2) The bright field images in Fig. 5f are very dim lacking contrast, it would be good to improve them.

Thank you for the kind suggestion. We have re-adjusted the figure brightness/contrast with imageJ to enhance contrast (also shown below). We have incorporated these changes into the revised manuscript (**Supplementary Fig. 5f**)

3) In the legend of Extended Fig. 6 sperm is repeated.

Thank you for pointing out this typo. The redundant “sperm” has been deleted.

4) Why are the currents reduced at positive potentials as $[Ca^{2+}]_i$ is increased? It would be good to determine if the reversal potential changes as $[Ca^{2+}]_i$ is increased. This is also the case also for the measurements presented as pH_i changes.

Thanks for your insightful comment. As you suggested, we calculated the reversal potential (E_{rev}) of CatSper under varying $[Ca^{2+}]_i$ in *Armh2^{+/-}* mouse sperm. The measured E_{rev} values remained largely consistent among different $[Ca^{2+}]_i$ (-1.92 ± 0.44 mV at 0 μ M, -1.30 ± 0.23 mV at 0.1 μ M, and -1.27 ± 0.75 mV at 10 μ M, respectively), as shown in the I-V curves of CatSper (Supplementary Fig. 7b). Thus, we think that the reduction of CatSper outward currents at positive potentials as $[Ca^{2+}]_i$ increased is unlikely due to the shift of reversal potentials.

To address this phenomenon, we propose the following explanation based on the ion selectivity of CatSper. Similar to other Ca^{2+} -selective channels, CatSper exhibits permeability to both monovalent (e.g., Na^+ , Cs^+) and divalent cations (e.g., Ca^{2+} , Ba^{2+}). Under physiological conditions, calcium channels exhibit Ca^{2+} -selectivity because of the much higher affinity of the Ca^{2+} with the ion-binding sites in the channel selectivity filter over monovalent ions, a hallmark characteristic of Ca^{2+} -selective channels (PMID: 12471162; Sather WA et al, *Annu Rev Physiol.* 2003). As a consequence, the currents

carried by divalent ions are usually small while the currents become much larger when using divalent ion-free recording solutions. This is the reason why Cs⁺-containing divalent ion-free (DVF) solutions were used to record CatSper currents (PMID: 16467839; Kirichok et al, Nature. 2006). Therefore, we infer that the decline in outward currents in response to elevated [Ca²⁺]_i arises from the competition of inside Ca²⁺ with Cs⁺ for the ion binding sites in the selectivity filter of the CatSper, and the competition exhibits as the blockage of Cs⁺-carried currents. Notably, the reduced currents at positive potentials in response to elevated [Ca²⁺]_i were also observed in a previous report (PMID: 31056283; Hwang JY et al, Cell. 2019).

On the other hand, compared to outward currents, CatSper-mediated inward currents (Cs⁺ influx) did not exhibit this [Ca²⁺]_i-dependent reduction because under negative potentials the inside Ca²⁺ will not enter into the selectivity filter thus there is no competition between Ca²⁺ and Cs⁺.

We wish that we had understood your question correctly and our explanation could relieve your concern.

5) How do the swimming parameters vary as a function of capacitation time in control vs Armh2 null? It seems 90 min is a fairly long time.

We thank the reviewer's insightful comment. Capacitation of mouse spermatozoa *in vitro* takes about 1.5-2 hours, with the peak reached around 1.5 hours (90 minutes). Therefore, 90 min is a typical time point used to perform motility analysis of sperm capacitation. To monitor the changes in sperm motility parameters during the capacitation process, we selected additional time points, including 0, 15, 30, 60, and 90 minutes, for CASA analysis in the revised manuscript (**Supplementary Fig. 8a**; also shown below).

The CASA results showed that the key hyperactivated motility parameters including curvilinear velocity (VCL) and amplitude of lateral head (ALH) exhibited a progressive increase over time in WT sperm, suggesting successful capacitation induction. In contrast, *Armh2*^{-/-} sperm showed no significant change in VCL or ALH during incubation in capacitation buffer (HTF), indicating a defect in hyperactivation and impaired capacitation ability.

6) It is unclear to me how the results in Fig. 6A show that hyperactivation is reduced in *Armh2*^{-/-} sperm. I cannot see that wild sperm hyperactivate, as the percentage of progressive motility does not change with 90 min capacitation. In contrast, the null sperm seem to hyperactivate, as their progressive motility is reduced. This contradicts Fig. 6b, where it is clearly seen that *Armh2*^{-/-} sperm do not hyperactivate but wild ones do?? The authors should show the values of hyperactivation.

We sincerely appreciate the reviewer's insightful observation regarding the hyperactivation analysis in Fig. 6a. We realized that the suboptimal CASA results might be attributed to the non-optimal conditions in our original capacitation buffer, which was a homemade BWW buffer. To address this, we repeated the CASA experiments using an optimized protocol (details provided in Methods) and by replacing the capacitation buffer with HTF medium. The new CASA results showed a decrease in progressive motility, and an increase in VCL and ALH in WT sperm after 90 min incubation in HTF medium, suggesting successful hyperactivation (Fig. 6a,b; also shown below). These results are also consistent with the results of beating waveform analysis. The updated CASA results have been included in the revised manuscript.

On the other hand, the reduction in progressive motility observed in *Armh2*^{-/-} sperm does not necessarily indicate hyperactivation, as other motility parameters of *Armh2*^{-/-}

sperm suggest defects in hyperactivation, but more likely result from impaired Ca^{2+} signaling. This phenomenon is consistent with previous studies showing that sperm from other CatSper component knockouts also exhibit decreased progressive motility after incubation in capacitation buffer (PMID: 17227845, Qi, H. et al, Proc Natl Acad Sci, 2007; PMID: 31056283, Hwang, J. Y. et al, Cell, 2019; PMID: 34998468, Hwang, J. Y. Cell Rep, 2022).

7) The details in Methods of the CASA experiments are really superficial. Why not measure directly hyperactivation in wild and mutant sperm. There are problems with these results.

We thank the reviewer's kind suggestion. As we mentioned at Comments 5) and 6), we optimized our protocol, reformed the CASA experiments and revised the methods of CASA in details. The revised version is as following:

Computer assisted sperm cell analysis (CASA)

Spermatozoa were isolated from the cauda epididymis and released into PBS buffer after a 10-minute incubation at 37 °C to allow sperm dispersion. The epididymal tissue was carefully removed using forceps, and the sperm-containing media was centrifuged at 200 g for 5 minutes. The resulting sperm pellet was resuspended in pre-warmed HTF medium (Sigma-Aldrich) and incubated in a tissue culture incubator at 37°C, 5% CO₂ for 10 minutes (designated as capacitation times of 0 minutes). Sperm motility was assessed at 0, 15, 30, 60, and 90 minutes of capacitation with a computer-assisted semen-analysis system (TOX IVOS II Automatic Sperm Analyzer, Hamilton Thorne Research). Briefly, a 30-μL aliquot of the sperm suspension was loaded onto a 100-μm-depth counting chamber (Leja) and observed under a negative phase-contrast field. Sperm cell tracks were recorded for 0.5 seconds at 60 Hz. For each specimen and time point, 100-150 spermatozoa were analyzed across ≥5 randomly selected fields to determine total sperm motility (%), progressive sperm motility (%), curvilinear velocity (VCL, μm/s), and amplitude of lateral head displacement (ALH, μm).

We thank this referee for their critical reading and constructive comments.

Referee #3:

In this manuscript by Zhao Q. et al the authors reveal that ARMH2 is a novel component of the CatSpermasome. The authors perform new cryo-EM analysis (building upon a previous publication) to improve the resolution of the cytosolic regions of CatSper, resulting in an overall map in the cytosolic region at $\sim 5\text{\AA}$ resolution. The placement of ARMH2 within this newly resolved map is supported by extensive Alphafold predictions, pulldown assays, and co-evolution analysis. Further experiments with CRISPR knockout mouse models and electrophysiology demonstrate that ARMH2 is a critical intracellular component of the CatSpermasome.

I approached this review focusing on the electron microscopy aspects of the manuscript, and will leave comments on the other sections (mouse models, electrophysiology, sperm motility) to experts in these particular areas. Although the obtained resolution in cytosolic map 2 is limited to $\sim 5\text{\AA}$, the clearly resolved secondary structure elements strongly support docking of an AlphaFold model of ARMH2. This is a great example of how even intermediate resolution cryo-EM maps from disordered and poorly resolved regions can be integrated with AlphaFold structure prediction and co-evolution analysis to gain important biological insights.

The electron microscopy experiments appear to be well performed, and support the primary conclusions of the manuscript. Overall, I am very supportive of publication of this manuscript. I have only relatively minor comments listed below...

We thank the referee for the invaluable time and effort in reviewing our manuscript and for acknowledging our work. Below, we provide a detailed, point-by-point response to the referee's comments, along with the corresponding revisions made to the text.

1. In the first paragraph of the results section, it is stated "To facilitate reliable component assignment in this region, we further optimized the cryo-EM sample". However, the methods section only states that "The isolated native CatSpermasome sample were prepared as previously reported", and the vitrification conditions appear to be the same as the previous publication. It is unclear to me what was actually optimized in the current manuscript? Were there any changes to the sample purification/vitrification protocol, or was it simply just a larger dataset? More details on exactly how the sample was "optimized" are needed.

We thank for the kind suggestion and apologize for potential confusion. The samples were optimized mainly in the following two aspects: 1) Except for the previously used lacey carbon grids, we primarily used Quantifoil 2 nm carbon film-coated grids in this study. The thinner carbon film provides better contrast, which we believe facilitates

higher-resolution imaging. 2) We also increased the size of the cryo-EM dataset to ensure a sufficient number of particles for improvement of the map quality in the cytosolic region.

We have revised the corresponding main text (in page 5) as follows: “To facilitate reliable component assignment in this region, we further optimized the cryo-EM sample preparation and data collection steps to improve the local map quality in the following aspects: 1) Except for the previously used lacey carbon grids, we primarily employed the 2 nm carbon film-coated grids for cryo-EM sample preparation, which provide better image contrast; 2) We collected a larger dataset (Supplementary Fig. 1).”

2. For particle picking, the methods and Extended Data Figure 1 only mention “iterative particle picking”. More details on how this picking was performed are warranted. Was this blob picking followed by template-based picking in CryoSPARC? Or was a more advanced particle-picking scheme, such as one of the neural-net based pickers employed?

We are sorry for the confusion. It was blob picking followed by template-based picking in CryoSPARC. We have revised Supplementary Fig. 1c and the corresponding Methods to make it clearer in the revised manuscript.

3. From Extended Data Figure 1 and the methods section it is not clear what data processing steps were performed in Relion versus CryoSPARC. I presume the iterative local classification was performed in Relion? But what about the signal subtraction step prior to classification, was this performed in CryoSPARC before exporting the subtracted particles for classification in Relion? Similarly, it is not clear whether or not the final focused refinement to obtain the 4.8Å map was performed in Relion or CryoSPARC. Please expand the methods section and/or Extended Data Figure 1 to include these details.

We apologize for the confusion. We initially attempted 3D classifications in both cryoSPARC and RELION, and ultimately used the results from CryoSPARC. Accordingly, all presented results were obtained using CryoSPARC. We have revised the corresponding Methods section as follows: “All structure determination procedures were conducted using cryoSPARC v4.”

4. Was 3D classification/refinement in Relion 5.0 performed with or without BLUSH regularization? If it was performed without, have the authors considered trying BLUSH regularization both in the local 3D classification and final refinement steps? For some

targets we have found BLUSH regularization to provide significantly improved resolutions and map interpretability, particularly on challenging small proteins.

We appreciate the referee's helpful suggestion. We attempted 3D classification and refinement in RELION5.0 with BLUSH regularization. However, we have consistently encountered issues of 'bad termination', for which we have not yet found an effective solution. As a result, we were unable to explore different parameters, such as angular step and searching angle. Among the few jobs that successfully completed, we have not observed improved results. A result of the 3D classification of the intracellular region is shown in the figure below (with BLUSH regularization, angular step 3.7°, searching angle 20°). At this stage, we are unable to ascertain whether RELION could yield better computational results. Therefore, we have opted to retain the results obtained using cryoSPARC in the manuscript.

5. Was the final refinement to obtain the improved resolution of cytosolic map-2 performed on the signal subtracted particles, or was the final refinement performed with the complete (non-subtracted) particles with just a mask around the cytosolic region?

Thank you for the question. The final refinement was performed on the signal subtracted particles. We also tried using the complete particle with a mask around the cytosolic region; however, the results were not superior to the currently presented map (shown below). We have updated Supplementary Fig. 1c and the corresponding method section to clarify this point.

Results of the refinements performed with complete particle with a mask around the cytosolic region (left) and the ARMH2-EFCAB9-CATSPERZ region (right)

Minor:

1. In Figure 1D it is a little hard to see the atomic model in the transparent volume. Maybe consider increasing the map transparency slightly, particularly for the ARMH2 subunit.

Thanks for the kind suggestion. We have increased the map transparency in Fig. 1d in the revised manuscript. The fitted model now can be clearly visualized.

2. Second to last sentence in the second paragraph of the results section states “The cryo-EM resolved regions of the ARMH2-EFCAB9-CATSPER ζ subcomplex are primarily in areas with high predicted local distance difference test (pLDDT) scores, supporting the assertion that the unassigned density belongs to ARMH2 (Fig. 1B)”. I believe this should point to Fig. 1D instead?

We sincerely appreciate the reviewer’s careful reading. It should indeed refer to Fig. 1d. We have corrected it accordingly.

3. “Alternatively, we transiently co-expressed ARMH2-3xFlag, Strep-EFCAB9, and CATSPER ζ -GFP in HEK293 cells and examined whether purification of one component would pull-down the other two proteins (Extended Data Fig. 2d).” I believe this should point to Main Figure 2D, not Extended Data Fig. 2D.

Thanks for pointing out this mistake. It should indeed point to Fig. 2d. We have corrected it accordingly.

4. “Immunoblotting analysis of the ARMH2-null sperm revealed that the detected transmembrane subunits, including CATSPER1, CATSPER β , δ , and θ , were all reduced to approximately half or less compared to WT, whereas CATSPER ζ was entirely absent (Fig. 4a and Extended Data Fig. 4b). There is no western blot or pulldown data in Extended Data Fig. 4b

The correct corresponding figure should be “Fig. 4a,b”, now “Fig. 4c,d” in the revised manuscript. We apologize for the oversight regarding the figure reference. We have thoroughly reviewed the manuscript to identify and rectify any other potential typographical errors.

5. “The cytosolic ternary subcomplex is likely involved in these intercellular junctions” – replace intercellular with intracellular

We appreciate your suggestion. “intercellular” has been replaced with “intracellular”.

6. Last word of Fig. 1 legend. “labelled” should be “labeled”

Thanks for pointing out this typo. We have amended it accordingly.

7. First sentence of Fig. 2a legend. “Predicated” should be predicted

Thanks for pointing out this typo and we have corrected it as suggested.

We thank this referee for their critical reading and constructive comments.

Referee #4:

This manuscript by Zhao et al. is a logical follow-up study to this group previous manuscript describing CatSper structure. In that work, the authors used a combination of cryo-EM and mass spectrometry of what they called the CatSpermasome. In addition to all CatSper subunits, several other proteins were found including Armh2. In the present manuscript, further evaluation of cryo-EM coupled to the analysis of predicted structures using AlphaFold 2 predicted ARMH2 to be part of a ternary complex with EFCAB9 and CATSPERζ. Consistent with this prediction: phylogenetic analysis showed co-evolution of Armh2 with other CatSper subunits and null mutant mice are severely subfertile. Moreover, sperm from the Armh2 KO are phenotypically similar to those of Efcab9 and Catsperz null mice. Interestingly, absence of Armh2 reduced the overall levels of the pore-forming CatSper channel subunits, disrupted CatSper structure and eliminated pH dependence of CatSper currents. Overall, this work is excellent and used a very complete set of state-of-the-art methodologies to study the CatSper complex in general and Armh2 in particular. I have only a few comments.

We sincerely appreciate the referee's time and effort in reviewing our manuscript and thank for the insightful comments. Below, we provide a detailed, point-by-point response to the referee's comments, along with the corresponding revisions made to the text.

Specific points

1. In mouse, Armh2 is known as 17000 16Grik. ARMH2 in Humans is also known as C6orf229. Armh2 is not well represented in mouse sperm proteomic databases. However, it can be found with the alternative name (Grahn et al. DOI: 10.1038/s41467-023-40855-0). This information should be included.

We thank the referee for the kind suggestion. We have added this information in the revised manuscript as follows (page 9): “We further performed a quantitative proteomic analysis to systematically evaluate the changes of the CatSper components between Armh2-null and WT sperm. ARMH2, also known as 1700016G14Rik in mice and C6orf229 in humans, can be identified in proteomic databases using these alternative names (Grahn et al. Nat Commun, 2023).…”

2. I was not able to find the quantitative proteomic table used for the generation of Fig. 4c. Please include an excel table as supplementary information comparing results from wild type and Armh2^{-/-} sperm proteomes. This table will be a very important resource for the field.

We thank the referee for the insightful suggestion. We have included this data as **Supplementary Data 1** (Excel format) in the revised manuscript.

3. In vitro fertilization measured by the percentage of two-cell is about 30 %. Do the authors know the percentage of these 2-cells that reached the blastocyst stage?

We sincerely appreciate the reviewer's insightful question regarding the developmental potential of two-cell embryos. Our previous *in vitro* fertilization (IVF) data suggest a successful fertilization rate of over 20% for *Armh2*^{-/-} sperm, as estimated by the percentage of 2-cell embryos. This appeared higher than the results from *in vivo* mating. To further investigate, we repeated the IVF experiment and extended the *in vitro* embryo culture to the blastocyst stage. We found that the rate of embryos development from the 2-cell stage to blastocyst is much lower in the *Armh2*^{-/-} group compared to the WT group (Fig. 6f, also shown below). These new findings suggest that the actual IVF successful fertilization rate of *Armh2*^{-/-} sperm is substantially lower than 20%, with most of the observed 2-cell embryos likely arising from parthenogenetic activation. These results align more closely with the *in vivo* mating results.

Response to Referees' Comments (NCOMMS-25-17935A):

Reviewer #1:

In my opinion, authors have addressed all reviewer's questions in a proper way and they have also included complementary assays to reinforce their results like new initial conditions for computer assisted sperm cell analysis experiments (CASA), Additional developmental analysis for the in vitro fertilization (IVF) experiments, or the addition of western and proteomic data. The results shown in this revised manuscript indicate that ARMH2 is crucial for sperm function and they provide new insights into the composition and functional regulation of CatSper.

Reviewer #2:

The authors have properly dealt with the majority of the suggestions and questions posed by the reviewers. There are a couple of Figs., like Supp. Fig. 8a where the Y axes should say Curvilinear and it says Curve linear. On line 322 of the revised manuscript the authors have repeated analysis, please correct.

I believe the paper is ready to be published.

Reviewer #3:

The authors have properly addressed all of my critiques. I am supportive of publishing this work without further revision.

Reviewer #4:

The authors have addressed all my queries.

We are pleased that all four reviewers are satisfied with our revisions and supportive of publication. In response to reviewer#2's suggestion, we have revised the typographical errors in the revised manuscript. We sincerely appreciate for the reviewers' continued efforts and valuable feedback in reviewing our manuscript.